# Organoid modeling reveals the tumorigenic potential of the alveolar progenitor cell state

Jingyun Li [ID][1,2,3,8✉], Susanna M Dang [ID][1,2,8], Shreoshi Sengupta[1,2,8], Paul Schurmann [ID][1,4], Antonella F M Dost [ID][1,2], Aaron L Moye[1,2], Maria F Trovero [ID][1,2], Sidrah Ahmed[1], Margherita Paschini[1,2], Preetida J Bhetariya[5], Roderick Bronson[6], Shannan J Ho Sui [ID][5] & Carla F Kim [ID][1,2,6,7✉]

## Abstract

**Cancers display cellular, genetic and epigenetic heterogeneity, complicating disease modeling. Multiple cell states defined by gene expression have been described in lung adenocarcinoma (LUAD). However, the functional contributions of cell state and the regulatory programs that control chromatin and gene expression in the early stages of tumor initiation are not well understood. Using single-cell RNA and ATAC sequencing in Kras/p53-driven tumor organoids, we identified two major cellular states: one more closely resembling alveolar type 2 (AT2) cells (SPC-high), and the other with epithelial-mesenchymal-transition (EMT)-associated gene expression (Hmga2-high). Each state exhibited distinct transcription factor networks, with SPC-high cells associated with TFs regulating AT2 fate and Hmga2-high cells enriched in Wnt- and NFκB-related TFs. CD44 was identified as a marker for the Hmga2-high state, enabling functional comparison of the two populations. Organoid assays and orthotopic transplantation revealed that SPC-high, CD44-negative cells exhibited higher tumorigenic potential within the lung microenvironment. These findings highlight the utility of organoids in understanding chromatin regulation in early tumorigenesis and identifying novel early-stage therapeutic targets in Kras-driven LUAD.**

**Keywords** Lung Cancer; Organoids; Kras; Alveolar; Cell State
**Subject Categories** Cancer; Respiratory System

## Introduction

Cell plasticity endows cancer cells the power to dynamically transition between different cell states, which can be defined by patterns of gene expression and chromatin accessibility, without gaining additional genetic alterations. Cell plasticity has been implicated in cancer initiation, progression, tumor heterogeneity, and drug resistance (Torborg et al, 2022). Targeting particular cellular states that contribute to cancer cell plasticity is held back by a lack of facile in vitro models that recapitulate these states and a limited understanding of the underlying mechanisms. It is not understood which cellular states are directly responsible for the functional properties of cancer cells, especially in the early stages of disease. Answering these questions is particularly critical for improving lung cancer patient survival since most lung cancers are diagnosed at advanced stages; thus, defining cell states present at early stages of lung cancer has promise to allow new methods to detect and intervene in advanced disease.

Mouse models, coupled with single-cell (sc) sequencing (seq) techniques, have facilitated the study of cell plasticity and how they contribute to tumor heterogeneity in lung cancer. Kras activation (20–30%) and Trp53 loss of function (50–70%) are common mutational events in human non-small cell lung cancer, particularly in lung adenocarcinoma (LUAD), the most common type of lung cancer in patients (Collisson et al, 2014; Gibbons et al, 2014). Activation of Kras alone (K) or together with P53 loss (KP) in genetically engineered mouse models (GEMM) are able to initiate clonal outgrowth of lung adenocarcinoma in vivo (DuPage et al, 2009). Compared to the K model, the tumor progression in the KP model is more rapid with a higher tumor burden, higher tumor grade, and more metastatic disease. Single cell RNAseq and ATACseq indicated that alveolar type 2 (AT2) cells, which are the predominant cell type of origin for lung adenocarcinoma in the KP mouse model, employ stereotyped programs during a time course of tumorigenesis (Kadur Lakshminarasimha Murthy et al, 2022; Marjanovic et al, 2020; LaFave et al, 2020; Ferone et al, 2020). After the onset of Kras activation and P53 loss (KP hereafter), cells demonstrated gene expression similar to AT2 cells. Next, many other signatures including a mixed phenotype of AT2 and alveolar type I (AT1) cells (AT1/AT2 mixed) and a gastric-like signature were upregulated, indicating that AT2 cells are experiencing lineage infidelity in response to oncogenic Kras. At later stages, tumor cells exhibited the expression of genes related to mesenchyme and loss of epithelial state (EMT-like). Lineage tracing coupled with chromatin

[1]Stem Cell Program, Divisions of Hematology/Oncology and Pulmonary Medicine, Department of Pediatrics, Boston Children's Hospital, Boston, MA 02115, USA. [2]Department of Genetics, Harvard Medical School, Boston, MA 02115, USA. [3]Department of Colorectal Surgery, Sir Run Run Shaw Hospital, Zhejiang University School of Medicine, Hangzhou 310016, China. [4]Department of Biology, Faculty of Science, Utrecht University, 3584 CH Utrecht, the Netherlands. [5]Harvard Chan Bioinformatics Core, Department of Biostatistics, Harvard T.H. Chan School of Public Health, Boston, MA 02115, USA. [6]Rodent Histopathology Core, Harvard Medical School, Boston, MA 02115, USA. [7]Harvard Stem Cell Institute, Cambridge, MA 02138, USA. [8]These authors contributed equally: Jingyun Li, Susanna M Dang, Shreoshi Sengupta. ✉E-mail: lijingyun@zju.edu.cn; carla.kim@childrens.harvard.edu

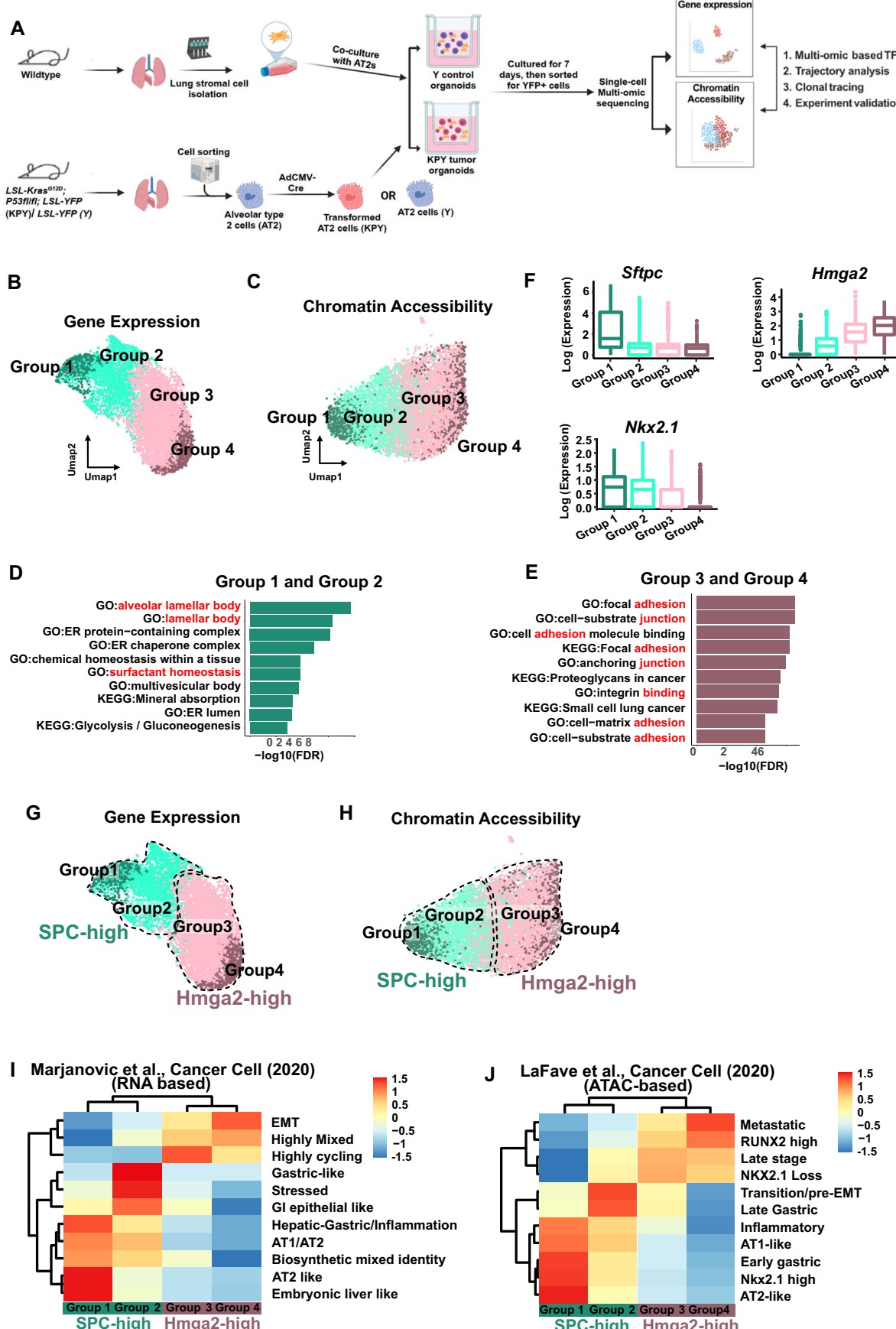

**Figure 1.  Two cell states exist in tumor organoids which are marked by unique gene expression signatures and chromatin accessibility signatures.**

(A) Experimental pipeline of using single cell multi-omic sequencing to analyze KPY tumor organoids. AT2, alveolar type 2 (AT2) cells. TF, Transcription factor. (B) Gene expression Umap showing the cell group identity. (C) Chromatin accessibility Umap showing the cell group identity. (D) Top 10 Gene ontology terms of genes highly expressed in Group 1 and Group 2. (E) Top 10 Gene ontology terms of genes highly expressed in Group 3 and Group 4. (F) Boxplots highlighting expression level (Log(TPX +1), color bar of Umaps and Y axis of boxplots) of selected genes. The central line represents the median, the box encompasses the interquartile range (IQR) (25th to 75th percentile), and the whiskers extend to the minimum and maximum values within 1.5 times the IQR. Outliers are shown as individual points beyond the whiskers. Group 1: $n = 773$; Group 2: $n = 2514$; Group 3: $n = 4367$; Group 4: $n = 769$. (G) Umap of gene expression assay showing the cell state definition in 7 days KPY organoids. (H) Umap of chromatin accessibility assay showing the cell state definition in 7 days KPY organoids. (I) Signature scores of gene programs (rows) (Marjanovic et al, 2020) in each cell state (columns). (J) Scores of chromatin co-accessibility modules (rows) (LaFave et al, 2020) in each cell state (columns).

analysis of lung tumorigenesis in KP mice further indicated that AT2 cells follow two non-overlapping evolutionary paths for tumor progression, with a gastric-endoderm state in one path and mixed lung cell states featured in the second path (Yang et al, 2022). These findings indicate that AT2 cells have at least two sets of regulatory programs for tumorigenesis in vivo in the KP GEMM.

Organoids are in vitro three-dimensional structures derived from normal progenitor cells or cancer cells. Unlike traditional in vitro models for tumor studies, including two-dimensional cancer cell lines and patient-derived xenograft models, organoids derived from patient samples have been verified to recapitulate the histopathology, molecular profiles, and response to therapies of their primary counterparts. Organoids derived from existing tumors, not from the normal cell of origin, have predominated the field. We recently established a tumor organoid system by co-culturing Kras activated AT2 cells with lung mesenchymal cells (Dost et al, 2020), making it possible to characterize the molecular changes that occur in the transition of normal epithelial cells to tumorigenic cells. Using this system, we showed that Kras tumor organoids can robustly recapitulate the key transcriptome changes of primary LUAD in GEMM and patient samples. After only seven days in culture, some cells in the Kras tumor organoids lose their original identity as AT2 cells, evidenced by decreased expression of AT2 cell marker genes and increased expression of lung developmental genes. We also observed heterogeneity in this system, but how the heterogeneity observed relates to key functions in tumorigenesis remained unknown.

To investigate the possible functional differences between cellular states in LUAD and to define their regulatory mechanisms, we used single-cell multi-omic sequencing to further characterize lung tumor organoids. Single-cell multi-omic sequencing (10X) allows simultaneous profiling of gene expression and chromatin accessibility from one single cell, which provides a more precise understanding of mechanisms contributing to cell identity and state transition. We used KP tumor organoids to allow us to determine how this system models the early and late stages of LUAD in GEMM. Multi-omic dissection on KP tumor organoids indicated that seven day tumor organoids recapitulate the tumor heterogeneity of primary lung adenocarcinoma both at the gene expression level and the chromatin accessibility level. KP tumor organoid cells could be separated into two cell states resembling the states defined in GEMM, one resembling AT2 cells and one similar to the EMT-like state (Hmga2-high). We used these gene expression and chromatin patterns to identify ways to separate and functionally compare cell states. These studies revealed that the AT2-like (SPC-high) state is tumorigenic in early-stage cancer and contributes to plasticity in lung tumorigenesis.

## Results

### Single-cell multi-omic profiling reveals two cell states exist in KP tumor organoids

We used single-cell multi-omic RNA and ATAC sequencing to dissect the epigenetic and transcriptomic responses of AT2 cells to oncogenic events, including KrasG12D mutation and P53 loss. Firstly, AT2 cells were sorted from Kras$^{LSL-G12D}$/P53$^{fl/fl}$/LSL-YFP (KPY) GEMMs using well-established surface markers (CD31-/CD45-/Epcam+/Sca1-) (Louie et al, 2022; Dost et al, 2020; Rowbotham et al, 2018; Fillmore et al, 2015; Lee et al, 2014). Kras activation and P53 loss were induced in the freshly sorted AT2 cells by infection with adenovirus 5 vector containing Cre recombinase driven by the ubiquitous cytomegalovirus (CMV) promoter (Ad5-CMV-Cre). Next, the Cre-induced AT2 cells were co-cultured with pre-expanded lung mesenchymal cells at an air-liquid interface for 7 days before harvest. We used 10X Genomics scMulti-omic sequencing to profile 7-day KPY tumor organoid cells and control YFP normal organoids, allowing gene expression and open chromatin profiling simultaneously from a single cell (Fig. 1A). The cells from KPY tumor organoids showed distinct gene expression patterns compared to YFP control organoids (Fig. EV1A–E). We subsequently focused on tumor cells and generated gene expression UMAP and chromatin accessibility UMAP separately using the scMulti-omic sequencing data (Fig. EV1F,G).

Next, we did cell clustering analysis using gene expression data and projected the cluster ID of each single cell onto the gene expression UMAP and chromatin accessibility UMAP (Fig. EV1F–H). We found that some clusters share a similar distribution on the chromatin accessibility UMAP even though they possess unique distributions on the gene expression UMAP (Fig. EV1H). We hypothesize that a cell state characterized by unique regulatory programs should possess unique signatures at both the gene expression and the chromatin accessibility layers. We merged the RNA clusters that share similar chromatin accessibility signatures to make sure that each cell state we defined possesses not only unique gene expression signatures but also unique chromatin accessibility signatures. With this strategy, we defined four cell states: Group 1 (Cluster 4), Group 2 (Cluster 2, 6, 7, and 9), Group 3 (Cluster 0, 1, 3, and 10), and Group 4 (Cluster 5 and 8) (Figs. 1B,C and EV1H). Although we identified four distinct cell states, Group 1 shared many highly expressed genes with Group 2, and Group 3 shared many highly expressed genes with Group 4 (Fig. EV1I). Genes highly expressed in Group 1 and Group 2 were enriched for alveolar lamellar body and surfactant homeostasis Gene Ontology (GO) terms, whereas genes differentially expressed in Group 3 and Group 4 were enriched in genes associated with cell

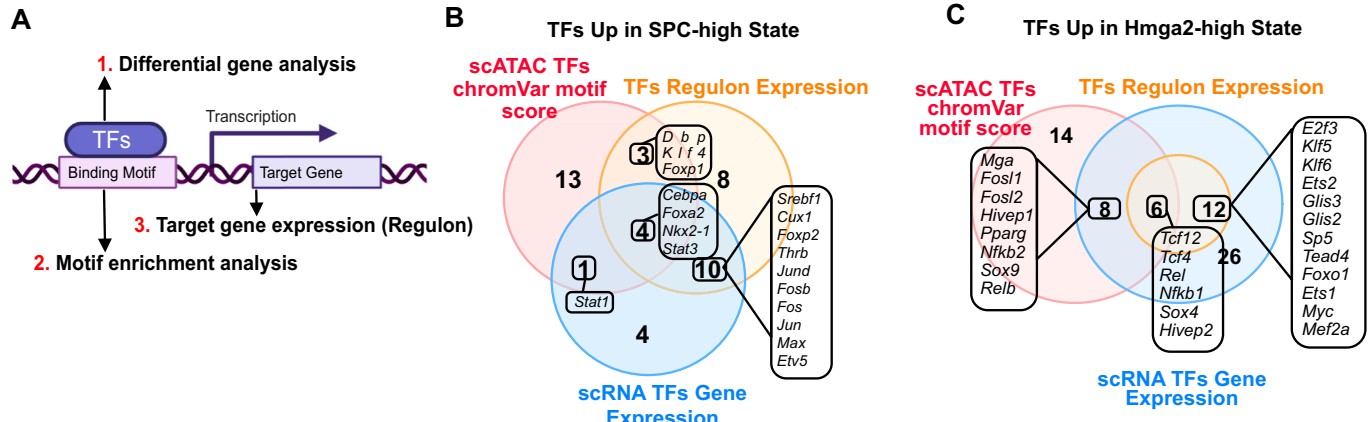

**A** 1. Differential gene analysis

2. Motif enrichment analysis

3. Target gene expression (Regulon)

**B** TFs Up in SPC-high State

scATAC TFs chromVar motif score

TFs Regulon Expression

scRNA TFs Gene Expression

**C** TFs Up in Hmga2-high State

scATAC TFs chromVar motif score

TFs Regulon Expression

scRNA TFs Gene Expression

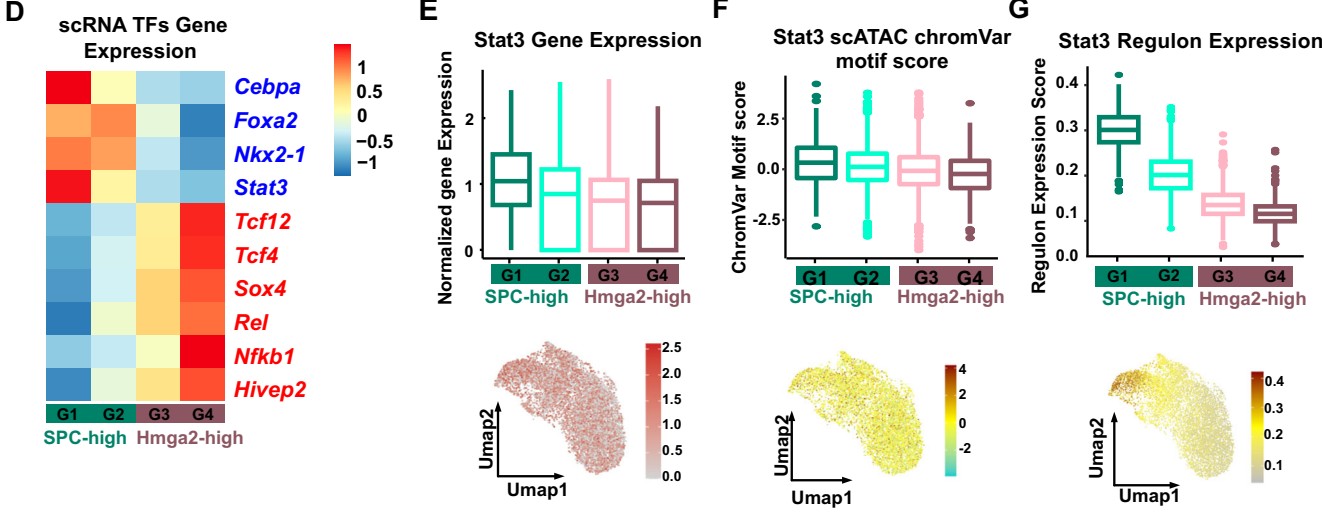

**D** scRNA TFs Gene Expression

**E** Stat3 Gene Expression

**F** Stat3 scATAC chromVar motif score

**G** Stat3 Regulon Expression

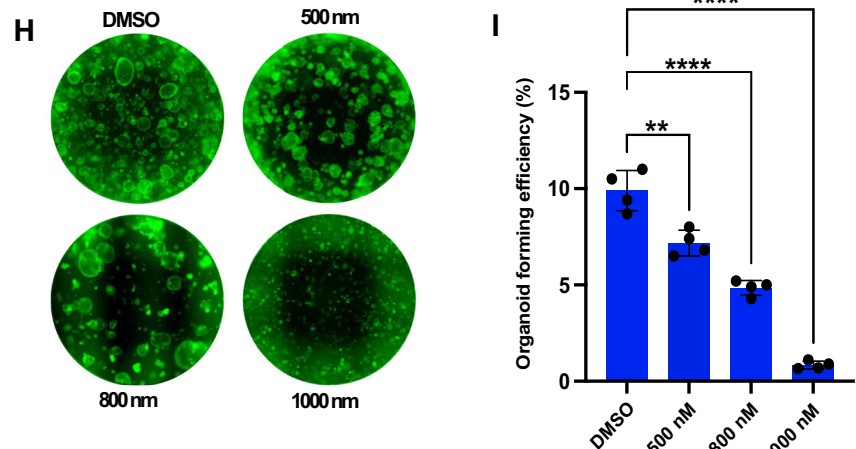

**H** DMSO  500 nm  800 nm  1000 nm

**I** Organoid forming efficiency (%)

**Figure 2. Using scMulti-omic data to dissect the regulatory programs underlying tumor heterogeneity.**

(A) Analysis strategy for identifying the transcription regulators underlying tumor heterogeneity. (B) Venn diagram showing the overlap of TFs enriched in SPC-high cells using differential gene analysis, chromVar motif score analysis and Regulon expression analysis. (C) Venn diagram showing the overlap of TFs enriched in Hmga2-high cells using differential gene analysis, chromVar motif score analysis and Regulon expression analysis. (D) Summary of all candidate regulators for SPC-high and Hmga2-high cells. The gene expression levels for each candidate regulator are shown using heatmaps. (E–G) Showing candidate regulator for Spc-high cells. The gene expression level (E), motif enrichment score (F), and regulon expression score (G) of Stat3 are shown both in boxplots and Umap. The central line represents the median, the box encompasses the interquartile range (IQR) (25th to 75th percentile), and the whiskers extend to the minimum and maximum values within 1.5 times the IQR. Outliers are shown as individual points beyond the whiskers. G1: $n = 773$; G2: $n = 2514$; G3: $n = 4367$; G4: $n = 769$. (H) Representative images showing transwells with organoids grown from DMSO- and Stattic-treated KPY cells in co-culture (magnification = 4×). (I) Quantification of organoids forming efficiency of DMSO- and Stattic-treated KPY cells in co-culture. Each dot indicates one biological replicate ($n = 4$). The data represents the mean ± SD. $p$-Value was calculated using an unpaired t-test with Welch's correction. $p$-Values = 0.0002, <0.0001, and <0.0001 (from bottom to top).

adhesion, cell junction, and cell-matrix adhesion, implying different cell functions (Fig. 1D,E).

Consistent with our GO analysis, AT2 marker *Sftpc* and lung lineage-specific transcription factor *Nkx2.1* showed higher expression in Group 1 and Group 2. Group 3 and Group 4 showed higher expression of *Hmga2*, which has been reported to regulate EMT programs during tumor progression (LaFave et al, 2020; Dost et al, 2020; Winslow et al, 2011; Naranjo et al, 2022) (Figs. 1F and EV1J). The patterns are also reminiscent of previous findings that lung adenocarcinoma has an altered expression of lung lineage specifiers and oncogenic signal pathways (Zewdu et al, 2021; Orstad et al, 2022; Snyder et al, 2013; Mollaoglu et al, 2018). Our analyses suggest that there are mainly two distinct cell states that exist in the tumor organoids: an SPC-high state resembling AT2 cells and lung lineage specifiers (Group 1 and Group 2) and an Hmga2-high state with similarity to known oncogenic signaling pathways (Group 3 and Group 4). The two groups within each major state possess unique chromatin accessibility signatures, indicating more complex heterogeneity (Fig. 1G,H). Herein we have largely focused on comparing the two major organoid cell states, SPC-high and Hmga2-high.

### Tumor organoids recapitulate the heterogeneity of in vivo tumors both at gene expression and chromatin accessibility level

Next, we wanted to know whether the cell states we observed in tumor organoids were reminiscent of those from in vivo tumors. We compared our dataset with two studies focused on primary lung adenocarcinoma with the same genetic background ($KRAS^{G12D}/P53^{Loss}$). Marjanovic et al did single-cell RNA-seq of in vivo lung adenocarcinoma at seven stages, from pre-neoplastic to adenocarcinoma, in the KPY model (Marjanovic et al, 2020). They defined 11 transcriptome programs to decipher the tumor heterogeneity during lung carcinogenesis. We checked the enrichment of those 11 transcriptome programs in the four cell states we identified in KPY organoids and found that the two subtypes within each cell state tended to cluster together based on their enrichment of those 11 transcriptome programs (Fig. 1I). The Hmga2-high organoid states are enriched for the EMT and highly mixed gene expression programs, with slight differences between Groups 3 and 4. Group 1 of the SPC-high tumor organoid state was highly enriched for the Marjanovic AT2-like and AT1/AT2 programs, whereas Group 2 was highly enriched for the gastric-like, stressed, and GI epithelial-like programs. Eleven chromatin co-accessibility modules were identified by LaFave et al, using scATAC-seq to characterize tumor heterogeneity at the chromatin

accessibility level from initiation stage to metastasis state in the KPY model (LaFave et al, 2020). The cell states we found in KPY tumor organoids also clustered together based on their enrichment of those 11 chromatin co-accessibility modules (Fig. 1J). Within the SPC-high cells, Group 1 most enriched AT2-like, Nkx2.1 high, early gastric, and AT1-like co-accessibility modules of LaFave et al, whereas Group 2 most enriched the late gastric and transition/Pre-EMT co-accessibility modules (Fig. 1J). Within the Hmga2-high cells, Group 3 most enriched the LaFave et al NKX2.1 loss and late-stage co-accessibility modules, while Group 4 most enriched metastatic and RUNX2 high co-accessibility modules. These results indicated that the KPY organoids at a single time point can faithfully recapitulate the major cell states of in vivo KPY lung adenocarcinoma both at the gene expression and at the chromatin accessibility level.

### ScMulti-omic data dissects the regulatory programs underlying tumor heterogeneity

Since the organoid system can recapitulate the heterogeneity of in vivo tumors at the gene expression level and chromatin accessibility level, we used the multi-omic data to identify the regulatory programs underlying tumor cell states. For each transcription factor, we compared the expression level, motif enrichment score, and target gene expression level between SPC-high cells and Hmga2-high cells (Fig. 2A). We identified the transcription factors for which all three layers of comparison were enriched in one specific cell state (Fig. 2B,C). For example, we found that the expression level of Nkx2.1 is higher in SPC-high cells, and the binding motif of Nkx2.1 is more enriched in the chromatin regions, which are more open in SPC-high cells. The target genes of Nkx2.1 were also upregulated in SPC-high cells (Fig. EV2A–C). The result is consistent with previous findings that NKX2.1 expression helps to keep lung lineage identity and loss of NKX2.1 correlates with the invasive phenotype (Orstad et al, 2022; Snyder et al, 2013; Camolotto et al, 2018). Nfkb1 showed opposite patterns and is a candidate regulator for Hmga2-high cells (Fig. EV2D–F).

With our data analysis strategy, four key potential regulators were identified for Spc-high cells, including Cebpa, Foxa2, Nkx2-1, and Stat3 (Fig. 2B,D,E–G). Previous studies have verified that Cebpa is a gatekeeper for normal AT2 identity (Cassel and Nord, 2003; Martis et al, 2006). Nkx2-1 and Foxa2 are co-expressed in normal AT2 cells, and they can also coordinately regulate lung cancer cell growth and identity in a context-specific manner (Orstad et al, 2022; Snyder et al, 2013; Camolotto et al, 2018; Little et al, 2021). Like Nkx2-1, STAT3 is enriched in AT2 cells and

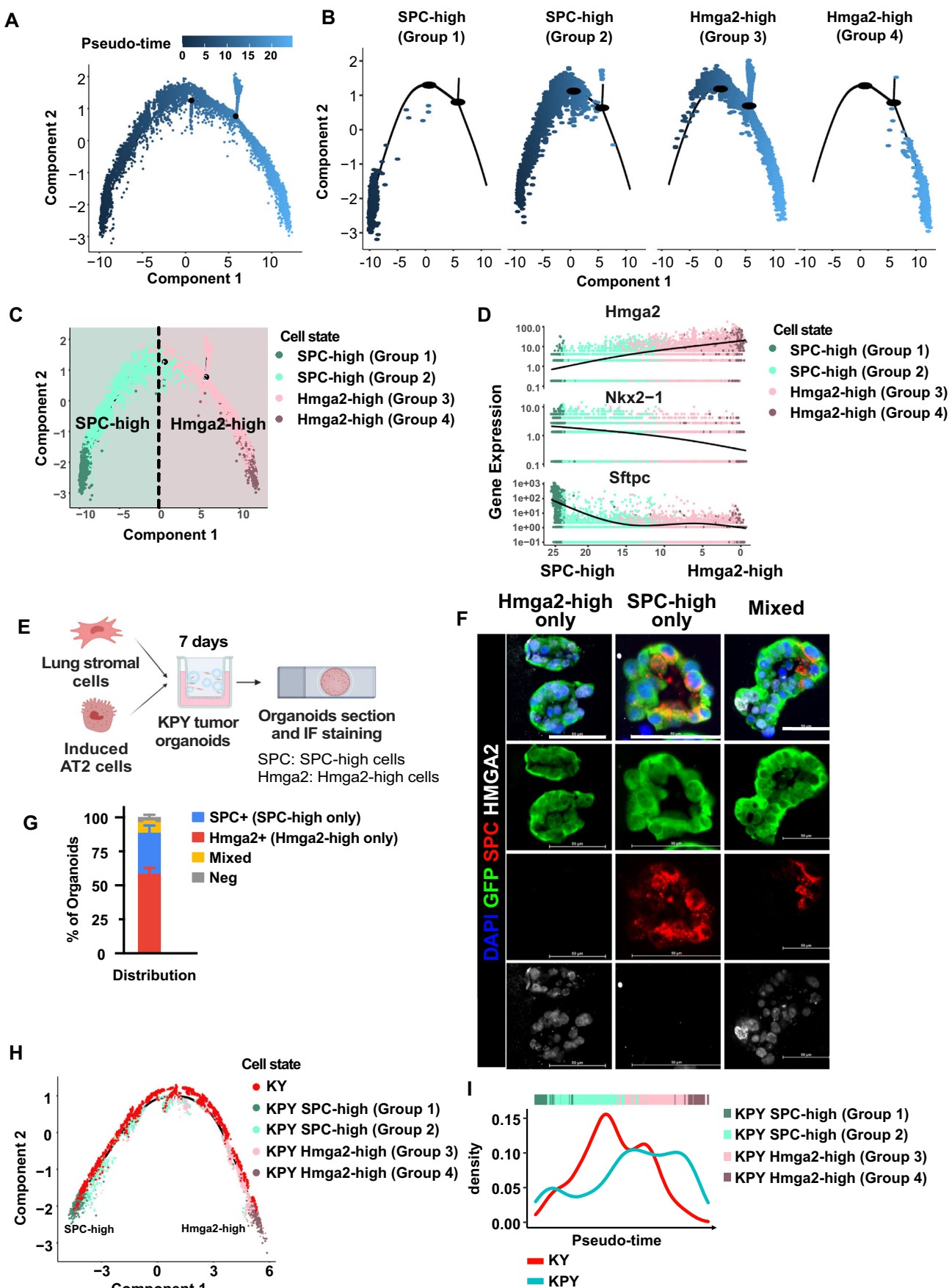

**Figure 3. SPC-high cells and Hmga2-high cells represent distinct path for tumorigenesis.**

(A) Monocle 3 pseudotime trajectory analysis of scMulti-omic sequencing expression data of KPY organoids. (B) Each cell state identified in Fig. 1G, H is illustrated in Monocle 3 pseudotime trajectory. (C) Monocle 3 pseudotime trajectory analysis of gene expression assay of single cells from KPY organoids. Cells are colored by cell state identity. (D) Gene expression level of selected genes along the pseudotime trajectory. Cells are colored by cell state identity. (E) Experiment strategy for identifying SPC-high cells and Hmga2-high cells in 7 days KPY organoids using IF staining. (SPC was used to label SPC-high cells; Hmga2 was used to label Hmga2-high cells) (F). Three types of KPY organoids, including Hmga2-high only (Hmga2+), SPC-high only (SPC +) and mixed (Hmga2 +/SPC+) based on immunofluorescence staining of SPC and Hmga2. (G) Quantification of the percentage of each type of organoids from (F). The data represents the mean ± SD ($n = 3$). (H) Monocle 3 pseudotime trajectory analysis of integrated data (see Methods) containing scMulti-omic sequencing expression data of KPY (Kras$^{G12D}$/P53$^{Loss}$/LSL-YFP) organoids and scRNA-seq data of KY (Kras$^{G12D}$/LSL-YFP)organoids. Cells from KY organoids are colored Red. Cells from KPY organoids are colored by cell identity defined in Fig. 1G, H. (I) Density distribution of cells from KPY or KY organoids along the pseudotime trajectory. Source data are available online for this figure.

STAT3 deletion had differential effects in vivo on Kras-driven tumorigenesis (Caetano et al, 2018; Mohrherr et al, 2013; Grabner et al, 2015; Zhou et al, 2015). Six potential regulators were enriched in Hmga2-high cells, including Tcf12, Tcf4, Sox4, Rel, Nfkb1 and Hivep2 (Figs. 2C,D, EV2D–F and EV2G,H). Tcf12 functions as a transcriptional suppressor for E-cadherin (Lee et al, 2012). Tcf4 has been reported to form complexes with β-Catenin to induce ZEB1, a key epithelial-to-mesenchymal transition activator (Peng et al, 2017; Liu et al, 2016; Shin et al, 2017). Sox4 was found to function as a positive regulator of the β-Catenin signal pathway by upregulating the expression of Tcf4 (Liu et al, 2016; Shin et al, 2017). Rel and Nfkb1 are members of the NFκB transcriptional complex (Karin et al, 2002). Hivep2 is a typical NF-κB inhibitor (Murphy et al, 2020; Roman et al, 2021), indicating that the NFκB signal pathway is dynamically regulated in the Hmga2-hi cells. NFκB signal pathway has been reported to evolve in tumor progression and metastasis (Chen et al, 2011). Notably, numerous studies have shown that Kras LUAD are dependent on Wnt signaling and NF-κB (Barbie et al, 2009).

Since STAT3 target genes were enriched in the SPC-high cell state, we further investigated the functional role of STAT3 using our organoid co-culture system. KPY tumor organoid co-cultures were treated with a Stat3 inhibitor (Stattic) after plating from day 7 to day 11. The results demonstrated that organoid formation was impacted in a dose-dependent manner (Fig. 2H,I). Moreover, the expression of putative Stat3 target genes Abca3 and Etv5 significantly decreased in Stattic-treated KPY tumor organoid cells compared to DMSO treatment (Fig. EV2I). This finding suggested that STAT3 function in the AT2-like state drives tumorigenesis in Kras mutant LUAD.

## SPC-high and Hmga2-high cells represent distinct paths for tumorigenesis

Next, we used Monocle, which is an unsupervised approach, to calculate a tumorigenesis pseudotime trajectory across all cell states as a means to begin to understand the relationships between KPY tumor organoid states. The trajectory analysis revealed the trajectory as a fork, indicating two different tumor evolution paths (Fig. 3A). Interestingly, the two groups of SPC-high cells were mainly distributed on the left branch while the two groups of Hmga2-high cells were mainly distributed on the right branch (Fig. 3B,C). The expression patterns of Hmga2, Nkx2.1, and Sftpc were checked to verify the cell state distribution on the tumorigenesis trajectory (Fig. 3D). The trajectory analysis suggests a gradual transcriptional transition within each state but the relationships between those two states are hard to infer since they are distributed on two branches. RNA velocity analysis suggests that there are limited transitions between SPC-high cells and Hmga2-high cells (Fig. EV3A,B).

To further define the cell states at the organoid level, we used immunofluorescence to detect markers of each state in KPY tumor organoid sections (Fig. 3E). We used antibodies for Sftpc and Hmga2 to identify the SPC-high and Hmga2-high cells, respectively. Three types of organoids were identified, including organoids only containing SPC+ cells (SPC +), organoids only containing Hmga2+ cells (Hmga2 +), and organoids comprised of SPC+ cells and Hmga2+ cells (SPC+ Hmga2 +) (Fig. 3F). We found that about 89% of the organoids are either SPC+ or Hmga2+, suggesting that after oncogenic Kras activation, most AT2 cells give rise to one of those two cell states (Fig. 3G). 7% of organoids possessed cells from both states (Fig. 3G).

Next, we determined whether the cell states we identified in the KPY organoids also exist in the Kras$^{G12D}$/LSL-YFP (KY) tumor organoids. We did scRNA-seq using 7 days of KY tumor organoids and then projected the cells onto the tumorigenesis trajectory of KPY tumor organoids (Fig. 3H). The density enrichment analysis of KY and KPY cells along the tumorigenesis trajectory indicate that KY organoids were more enriched in SPC-high cells than were KPY tumor organoids (Fig. 3I). Interestingly, we found that the K tumor organoids are almost depleted of the Hmga2-high Group 4, raising the possibility that Kras$^{G12D}$ mutation alone cannot drive AT2 cells to acquire the Hmga2-high Group 4 state, at least at the 7 day timepoint. This is consistent with previous findings that P53 loss can promote EMT-program activation (Powell et al, 2014; Semenov et al, 2022). These results may suggest that tumors with different oncogenic genotypes may differ in cell state composition, which may impact phenotypic variations in tumorigenesis based on genotype.

## SPC-high cells have enhanced organoid-forming ability in the presence of lung mesenchyme

We next sought to use cell surface markers to separate tumor organoid cells based on cell state features to compare the functional differences between the two cell states. Using differential gene analysis between the cell states, we found that CD44 can be used to distinguish SPC-high cells from Hmga2-high cells in 7 days KPY tumor organoids (Fig. 4A,B). IF staining indicated that CD44-positive cells are Hmga2+ and Spc-, supporting the idea of using CD44 to subset tumor organoid cells (Fig. 4C). Importantly, CD44 staining did not reveal distinct populations of cells in control organoids without Kras activation or from freshly isolated AT2 cells (Fig. EV4A,B). Next, we used FACS to subset cells from 7 days

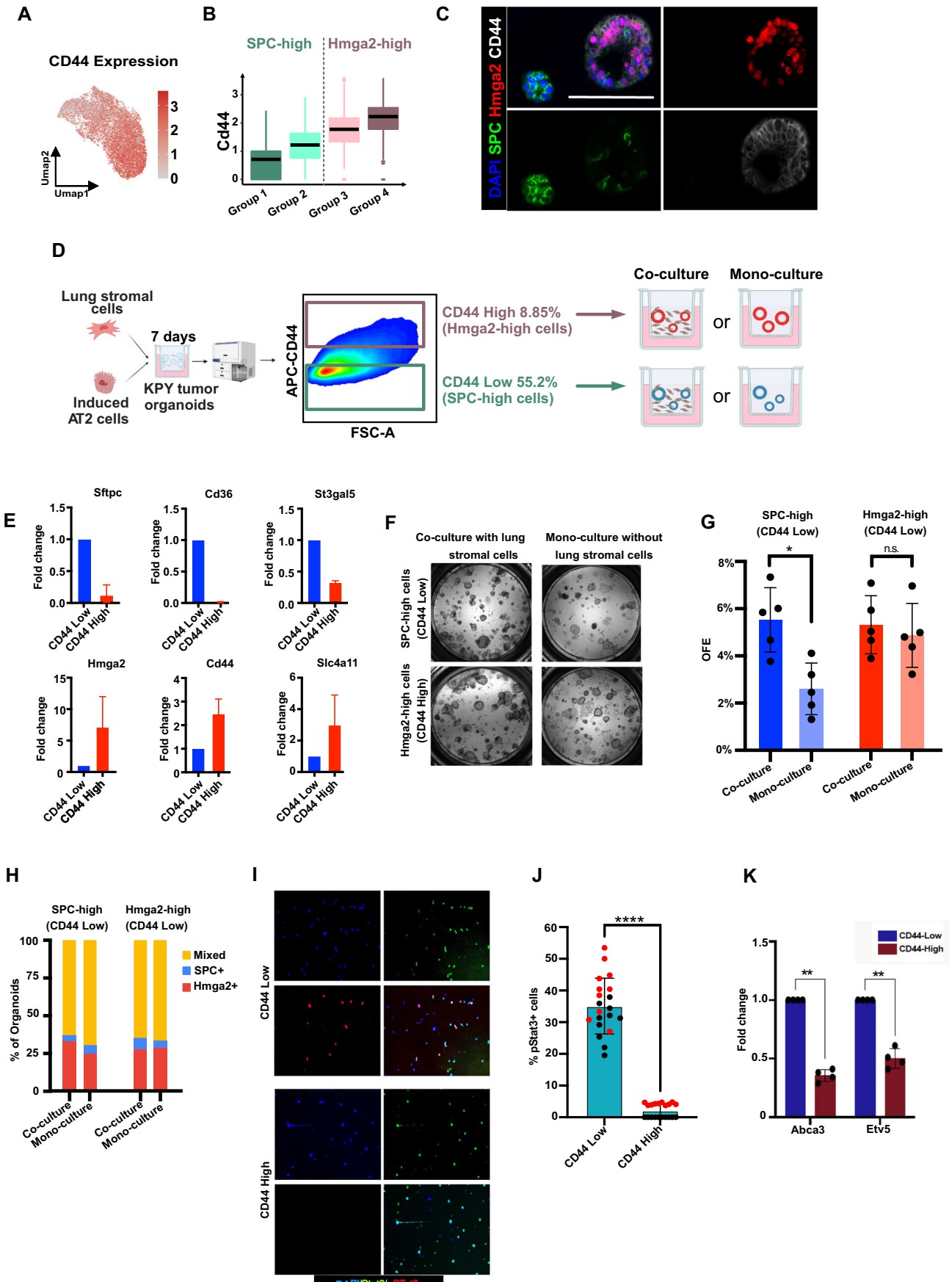

◄  **Figure 4.   Co-culture with lung mesenchymal cells enhance the organoids forming ability of SPC-high cells but not Hmga2-high cells.**

(A, B) Gene expression level of CD44 on Umap (A) and in four cell states (B). The central line represents the median, the box encompasses the interquartile range (IQR) (25th to 75th percentile), and the whiskers extend to the minimum and maximum values within 1.5 times the IQR. Outliers are shown as individual points beyond the whiskers. G1: $n = 773$; G2: $n = 2514$; G3: $n = 4367$; G4: $n = 769$. (C) IF staining indicates that CD44 co-stains with Hmga2. Scale bar, 100 μm. (D). FACS strategy to subset 7 days KPY organoids using CD44, then the CD44-high and CD44-neg population were cultured into organoids in co-culture or mono-culture condition. (E) qPCR showing that CD44 negative population expresses higher SPC-high lineage markers (Sftpc, Cd36, and St3gal5) and CD44 high population expresses higher Hmga2-high lineage markers (Hmga2, Cd44, and Slc4a11), the data represents the mean ± SD, sample size is 5 for each group. (F) Representative images showing organoids grown from SPC-high cells and Hmga2-high cells in Co-culture and Mono-culture conditions. (G) Summary of organoids forming efficiency of SPC-high cells and Hmga2-high cells in co-culture and mono-culture conditions. The data represents the mean ± SD, each dot indicates one biological replicate. Paired two-tailed t-test was performed, from left to right *p-Value = 0.034. N.s. p-Value = 0.43. (H) Quantification of the percentages of SPC+, Hmga2+ and mixed (Hmga2 + /SPC +) organoids in passage 1 organoids derived from SPC-high (CD44-neg) and Hmga2-high (CD44-high) cells in co-culture and mono-culture conditions. (I) Representative images of immunofluorescence staining of Stat3 and phospho-Stat3 in cytospin performed on CD44-Low vs. CD44-High cells (magnification = 4×, scale = 200 μm). (J) Quantification of phospho-Stat3 positive cells in CD44-Low vs. CD44-High cells. Each dot represents a field and the color of the dots (red and black) indicates two individual experiments ($n = 2$). p-Value was calculated using an unpaired t-test with Welch's correction. p-Value = <0.0001. (K) qPCR showing increased expression of putative Stat3 target genes Abca3 and Etv5 in CD44-Low KPY organoid cells, compared to CD44-High cells. The data represents the mean ± SD ($n = 4$). p-value was calculated using an unpaired t-test with Welch's correction. p-Values = 0.0043 and 0.0074 (from left to right). Source data are available online for this figure.

KPY organoids into CD44-high and CD44-low populations (Fig. 4D). qPCR analysis indicated that CD44-high populations have higher expression of genes highly expressed in the Hmga2-high state, including CD44, Hmga2, and Slc4a1, whereas the CD44-low population enriched for cells with gene expression similar to the SPC-high state, including Sftpc, Cd36, and St3gal4 (Fig. 4E). These results supported the rationale of using CD44 to subset and compare the KPY tumor organoid cell states.

Next, we investigated the organoid-forming potential of the two cell populations separated by CD44, approximating the two major cell states. After subsetting 7 days KPY tumor organoids into CD44-high (enriched for Hmga2-high cells) and CD44-low (enriched for SPC-high cells) populations, we cultured those cells into organoids with (co-culture) or without (mono-culture) lung mesenchymal cells to compare the organoid forming efficiency (OFE) (Fig. 4D). We found that both populations can grow tumor organoids in these conditions (Fig. 4F,G). The OFE of CD44-high cells was not affected by the existence of mesenchymal cells, but the OFE of CD44-low in mono-culture was significantly lower than that in co-culture with mesenchymal cells (Fig. 4F,G). We also found that a higher mesenchymal:tumor cell ratio correlates with a higher percentage of SPC+ organoids, suggesting that mesenchymal cells may support the survival of induced AT2 cells that follow the SPC-high cell state (Fig. EV4C,D). To investigate whether those two cell states can maintain their identity after passaging, we quantified the types of organoids derived from the CD44-sorted populations in the two conditions (Fig. 4H). Both CD44-low and -high cells could give rise to SPC +, Hmga2 +, SPC + /Hmga2+ organoids, regardless of whether they are cultured with or without lung mesenchymal cells, indicating that there may be plasticity between the cell states (Figs. 4H and EV4E,F).

To further elucidate the molecular effectors underlying the functional differences between the two tumor cell states, we focused on investigating the role of STAT3, a key regulator enriched in Spc-high/CD44-low cells. Immunofluorescence imaging from cytospin assays revealed that while both CD44-low and CD44-high populations exhibited comparable levels of total STAT3 protein, the CD44-low population showed a significantly higher percentage of phosphorylated STAT3 (pStat3) (Fig. 4I,J). Furthermore, qPCR analysis demonstrated increased expression of putative Stat3 target genes in CD44-low cells relative to CD44-high cells (Fig. 4K). These results strengthen our previous findings that, despite minor

differences in Stat3 mRNA levels between the SPC-high and Hmga2-high groups (Figs. 2A,B,D, EV2G,H, and 2E), notable differences exist in Stat3 activity and regulon expression between the two populations (Fig. 2G). Specifically, the higher expression of putative Stat3 target genes and the increased pStat3 levels in the Spc-high/CD44-low cells point to elevated Stat3 activity in this population. Together, these data suggest that the heightened Stat3 activity in CD44-low cells may serve as a crucial driver of the functional differences observed between CD44-high and CD44-low cells.

## SPC-high cells exhibit higher tumorigenic capacity in vivo compared to Hmga2-high cells

We hypothesized that the SPC-high cell state has enhanced tumorigenic capacity in the lung microenvironment in vivo, therefore we employed an orthotopic transplantation assay to compare the tumorigenic capacity of the KPY organoid cell states. After sorting CD44-high and CD44-low populations from 7 days KPY organoids, we transplanted the same number of cells from each population by intratracheal instillation as described (Louie et al, 2022; Dost et al, 2020) (Fig. 5A), and the mice were euthanized two months after transplantation to quantify the tumor burden. The mice that received CD44-low cells had a significantly higher tumor burden than CD44-high cell recipient mice (Figs. 5B and EV5A). The CD44-low recipient mice had significantly higher grade tumors (Grade III and IV) than CD44-high cell recipient mice (Fig. 5C,D). We also performed immunofluorescence to infer the cell state potential of organoid cells after transplantation. We found that CD44-high cell recipient mice had a higher percentage of Mixed tumors (SPC + /Hmga2 + ) whereas the CD44-low cell recipient mice had a higher percentage of SPC+ tumors (Figs. 5E and EV5B). To validate Kras and p53 allele recombination efficiency in both CD44-low and CD44-high KPY organoid cells, we extracted gDNA from CD44-low and CD44-high cells from KPY organoids at day 7 and performed PCR to amplify WT and mutant Kras and p53 alleles. We only observed PCR bands corresponding to the recombined Kras or p53 alleles in both the CD44-low and CD44-high samples, indicating that Cre-recombination at both the Kras and p53 alleles has occurred in all cells (Fig. EV5C). We also employed an orthotopic transplantation assay of Kras$^{LSL-G12D}$/P53$^{fl/fl}$/LSL-TdTomato (KPT) organoid

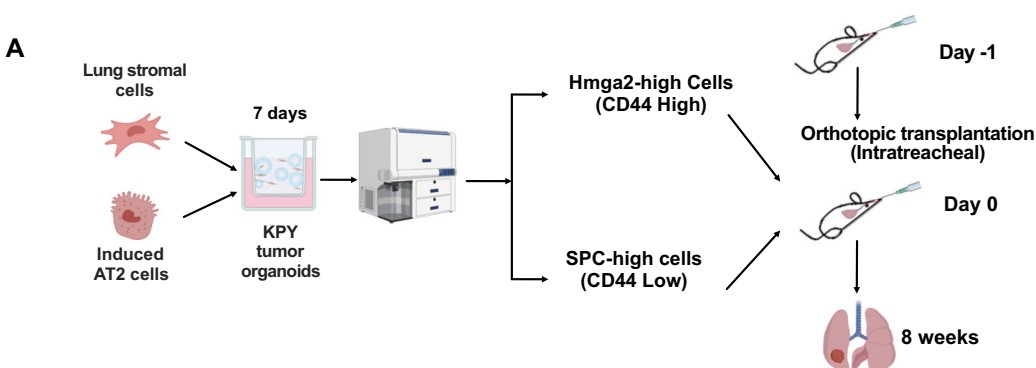

**A**

Lung stromal cells

Induced AT2 cells

7 days

KPY tumor organoids

Hmga2-high Cells (CD44 High)

SPC-high cells (CD44 Low)

Bleomycin Injury

Day -1

Orthotopic transplantation (Intratreacheal)

Day 0

8 weeks

**B**

Tumor burden (Lesion area/total area)

**

Control    CD44 Low    CD44 High

**C**

| | Grade I | Grade II | Grade III | Grade IV |
|---|---|---|---|---|
| CD44 Low | | | | |
| CD44 High | | | | |

**D**

% Tumors Per animal

Grade I    Grade III
Grade II    Grade IV

CD44 Low    CD44 High

**E**

Proportion of tumors per animal

SPC Only    Mixed
HMGA2 Only    Negative

n.s.
n.s.

CD44 Low    CD44 High

**F**

Alveolar type 2 cells

Kras^G12D P53^Loss

Lung Mesenchymal cells

SPC-high

Hmga2-high

Group 1    Group 2    Group 3    Group 4

Tumorigenic capacity in vivo

AT2 Identity

Stat3 signaling

Surface marker CD44

Wnt signaling

Nfkb signaling

**Figure 5. SPC-high cells have higher tumorigenic capacity than Hmga2-high cells in vivo.**

(A) Experiment strategy for subsetting Hmga2-high cells and SPC-high cells and evaluating their ability to contribute to tumors in vivo using Orthotopic transplantation assay (see Methods). (B) Quantification of tumor burden in CD44-High and CD44-Neg recipient mice. Paired two-tailed t-test was performed, **$p$-Value = 0.009, the data represents the mean ± SD, each dot indicates one biological replicate. (C) HE staining shows representative tumors at different grades from CD44-Neg recipient mice and CD44-High recipient mice. Scale bar, 100 μm. (D) Quantification of the percentage of tumor at different grades in CD44-Neg and CD44-High recipient mice. The data represents the mean ± SD, paired two-tailed t-test was performed for significant analysis, *$p$-Value = 0.016 (Grade III), 0.047 (Grade II), 0.033 (Grade I). (E) Quantification of the percentage of each tumor type (SPC only, Hmga2 only, and mixed) in CD44-Neg, and CD44-High recipient mice. The data represents the mean ± SD, paired two-tailed t-test was performed for significant analysis, *$p$-Value = 0.076 (Mixed), 0.053 (SPC-only). (F) Model for oncogenic changes in AT2 cells after the onset of Kras activation and P53 Loss. Source data are available online for this figure.

cells to compare the efficiency of transplanting CD44-low and CD44-high cells. Flow-cytometric analysis revealed equal proportions of EpCam+TdTomato+ cells from CD44-low and CD44-high transplants (Fig. EV5D), suggesting that the differential ability of the CD44-low and CD44-high cells is not solely due to differences in transplantation. Overall, the organoid and orthotopic assays indicated that the SPC-high cell state has a higher tumorigenic capacity in the lung microenvironment.

## Discussion

Our murine tumor organoid system faithfully models lung cancer cell states at both the gene expression level and the chromatin accessibility level, providing a means to identify and test key regulators of different tumor cell states. By combining single cell multi-omic, pseudotime analysis, and organoid functional studies, our findings are consistent with the phylogenetic relationships proposed by others using genetically engineered mouse models, and further support that the organoid system recapitulates the common tumor evolution path of its in vivo counterpart. Yang et al combined lineage tracing and transcriptome profiling to infer the phylogenetic relationship between cell states in the KPY model (Yang et al, 2022). We further describe the changes in AT2 cells after the onset of Kras activation and p53 loss, in which AT2 can take one of two paths for tumorigenesis. The molecular and functional differences of cells on those two paths from our study are summarized in Fig. 5F. Our results demonstrate many of the tumorigenic properties of the two cell states, particularly highlighting the capacity of SPC-high cell states, which most resemble AT2 cells, to drive tumorigenesis.

Our results demonstrate the utility of using organoid systems to dissect the distinct functional properties of cell states that contribute to tumor heterogeneity. Our previous study indicated that the tumor organoid system can model the early response of AT2 cells to Kras activation by comparing tumor organoids to control normal organoids (Dost et al, 2020). In that particular study, our organoid studies revealed that the loss of AT2 identity in LUAD occurs earlier than previously understood from in vivo studies. In Dost et al, we did not test the function of cells with decreased AT2 gene expression, and notably we only examined Kras-G12D, p53-wildtype (KY) cells at the single cell level. Here we characterize the gene expression and chromatin accessibility of Kras-G12D cells with p53 loss (KPY). Furthermore, here we directly compared the cell states we defined in KY and KPY backgrounds. KY tumor organoids had a higher percentage of SPC-high cells compared to KPY tumor organoids and do not exhibit

the evidence of Hmga2-high Group 4 cell state. The induced lung adenocarcinoma in Kras$^{G12D}$ mice grows much slower and shows lower metastasis frequency than tumors from Kras$^{G12D}$/P53$^{Loss}$ mice (DuPage et al, 2009). It is intriguing to hypothesize that these differences in cell state composition may explain the difference in phenotypic characteristics, such as metastatic potential, between the KY and KPY models. Future studies to compare the metastatic capacity of the cell states will be needed to address this hypothesis.

Our organoid forming assay and transplantation experiments showed that SPC-high, CD44-low cells exhibit increased tumorigenic capacity in the presence of niche cells in the lung. In organoid cultures, lung mesenchymal cells enhanced the growth of SPC-high cells, but did not impact Hmga2-high cells, consistent with the idea that the lung microenvironment supports the SPC-high cell state. By increasing the proportion of adjacent stromal cells within the co-culture system, we observed a notable increase in the population of SPC+ tumor cells, accompanied by a corresponding decrease in Hmga2+ tumor cells. Furthermore, whereas CD44-low and CD44-high cells exhibited similar transplantation efficiency, CD44-low cells had a significantly higher ability to initiate tumors in vivo. Consequently, we postulate that in the transplantation model, the tumorigenic function of SPC-high cells is supported by lung stromal cells; the identity of such cells remains to be determined. While numerous studies have highlighted CD44 as a marker gene associated with cancer stem cells, thus contributing to enhanced tumorigenic capacity, it is essential to note that the majority of these investigations primarily focus on later stages of tumor development, whereas our study specifically addresses the early tumor initiation stage.

Our findings substantiate the notion that tumor cells retaining a higher degree of original cell identity are likely accountable for tumor expansion during the initiation phase, as they are more adept at exploiting the surrounding microenvironmental cells for their benefit. We employed a well-established FACS strategy to selectively enrich AT2 cells, recognized as a predominant originating cell type for LUAD, to study the impact of oncogenic Kras at early time points. In this study, we have not compared tumor organoid development from other possible cells of origin, such as bronchioalveolar stem cells (BASCs). The exploration of whether distinct cell origins contribute to the observed tumor heterogeneity represents an intriguing avenue for further investigation. Additionally, we have not determined whether both cell states are required for tumor progression through paracrine signaling; the SPC-high state may rely on the HMGA2-high state or vice versa. Finally, our results suggest that SPC-high cells exhibit considerable plasticity and possess the potential to give rise to CD44-high cells during tumor progression. It remains possible that switching

between cell states occurs in later stages and/or metastasis; these aspects have not been studied here.

Our murine tumor organoid system and associated data provides a means to identify and probe the critical regulators of cell state and plasticity that drive human lung cancer. Our work shows that more emphasis is needed to understand and target the dependencies of AT2-like tumor cells, that is those cells that most closely resemble their normal counterparts. Many of the current targets being pursued based on the literature and studies of advanced lung cancer are focused on the pathways we found are active in the tumor organoid Hmga2-high state, such as the Wnt and Nfkb pathways. Our study suggests that interrupting these signaling pathways may not target the AT2-like SPC-high cell state, which may depend on distinct pathways including STAT3 activity. Furthermore, SPC-high cells produced organoids with Hmga2-high cells after passaging in culture or transplantation in vivo and vice versa, suggestive of plasticity between the tumor organoid cell states. Our work suggests that combination therapy targeting both cell states may be needed for more effective treatment of advanced lung cancer.

## Methods

### Reagents and tools table

| Reagent/Resource | Reference or Source | Identifier or Catalog Number |
| --- | --- | --- |
| **Experimental models** | | |
| Kras$^{LSL-G12D/+}$ | Jackson et al, 2001 | N/A |
| Kras$^{LSL\,G12D/+}$; p53$^{fl/fl}$ | Jackson et al, 2005 | N/A |
| Hsd:Athymic Nude-Foxn1nu | ENVIGO | Cat#6903F |
| **Recombinant DNA** | | |
| Ad5CMVempty | Viral Vector Core University of Iowa | Lot:Ad4154; Cat#VVC-U of Iowa-272 |
| Ad5CMVCre | Viral Vector Core University of Iowa | Lot: Ad4117; Cat#VVC-U of Iowa-5 |
| **Antibodies** | | |
| Rat monoclonal anti-CD45 APC [30-F11, BD] | BioLegend | Cat#BDB559864 |
| Rat monoclonal anti-CD31 APC [MEC 13.3, BD] | BioLegend | Cat# BDB551262 |
| Rat monoclonal anti-CD326 (EP-CAM) PE/Cy7 [G8.8] | BioLegend | RRID:AB_1236471; Cat#118216 |
| Rat monoclonal anti-Ly-6A/E (Sca1) APC/Cy7 [D7] | BioLegend | RRID:AB_1727552; Cat#560654 |
| CD44 Monoclonal Antibody (IM7) | Thermo Scientific | Cat#17-0441-82 |
| Rabbit monoclonal anti-SP-C [EPR19839] | Abcam | Cat#ab211326 |
| Rabbit monoclonal anti-TTF1 (Nkx2-1) [8G7G3/1] | Abcam | RRID:AB_1310784; Cat#ab76013 |
| Rabbit monoclonal anti-Phospho Stat3 (Tyr705) (D3A7) XP | Cell Signaling Technology | RRID:AB_2491009 Cat#9145 |
| Mouse monoclonal anti-Stat3 (124H6) | Cell Signaling Technology | RRID:AB_331757; Cat#ab9139 |
| Mouse monoclonal anti-Hmga2 [GT763] | GeneTex | Cat#GTX629478 |
| Goat polyclonal anti-GFP (YFP) | Abcam | RRID:AB_305643; Cat#ab6673 |
| Donkey anti-rat Alexa 594 | Invitrogen | RRID:AB_2535795; Cat#A-21209 |

| Reagent/Resource | Reference or Source | Identifier or Catalog Number |
| --- | --- | --- |
| Donkey anti-goat Alexa Fluor 488 | Invitrogen | RRID:AB_2534102; Cat#A-11055 |
| Donkey anti-goat Alexa Fluor 647 | Invitrogen | RRID:AB_141844; Cat#A-21447 |
| Donkey anti-rabbit Alexa Fluor 488 | Invitrogen | RRID:AB_141708; Cat#A-21206 |
| Donkey anti-rabbit Alexa Fluor 594 | Invitrogen | RRID:AB_141637; Cat#A-21207 |
| Donkey anti-mouse Alexa Fluor 647 | Invitrogen | RRID:AB_162542; Cat#A-31571 |
| **Oligonucleotides and other sequence-based reagents** | | |
| Kras-primer-1 | GTCTTTCCCCAGCA CAGTGC | |
| Kras-primer-2 | CTCTTGCCTACGCCA CCAGCTC | |
| Kras-primer-3 | AGCTAGCCACCATGG CTTGAGTAAGTCTGCA | |
| P53-primer-A | CACAAAAACAGGTTA AACCCAG | |
| P53-primer-B | AGCACATAGGAGGC AGAGAC | |
| P53-primer-D | GAAGACAGAAAAGG GGAGGG | |
| **Chemicals, Enzymes and other reagents** | | |
| GFR Matrigel | Corning | Cat#356231 |
| Bleomycin Sulfate | Sigma-Aldrich | Cat#B2434 |
| Dispase | Corning | Cat#CB-40235 |
| Collagenase/Dispase | Roche | Cat#10269638001 |
| DNase | Sigma-Aldrich | Cat#D4527 |
| Growth Factor Reduced Matrigel (3D Matrigel) | Corning | Cat# 356230 |
| Hyclone Fetal Bovine Serum (characterized; FBS) | GE Healthcare Life Sciences | Cat# SH30071.03 |
| 0.05% Trypsin-EDTA | GIBCO | Cat# 25-300-062 |
| Dimethyl Sulfoxide (DMSO) | Sigma-Aldrich | Cat# D2650 |
| BSA 7.5% Stock | Thermo Fisher Scientific | Cat# 15260037 |
| N-(2-Hydroxyethyl) piperazine-N'-(2-ethanesulfonic acid) Solution (HEPES) | Sigma-Aldrich | Cat# H0887 |
| **Software** | | |
| ImageJ | Schneider et al, 2012 | https://imagej.nih.gov/ij/ |
| GraphPad Prism for MacOS version 8.2.1 | GraphPad Software | https://www.graphpad.com/scientific-software/prism/ |
| FlowJo version 10.5.3 | Becton, Dickinson & Company | https://www.flowjo.com/ |
| Velocyto 0.17.17 | La Manno et al, 2018 | https://github.com/velocyto-team/velocyto.py |
| scVelo 0.1.25 | Theis lab | https://github.com/theislab/scvelo |
| CellRanger 2.0.1 | 10X Genomics | https://support.10xgenomics.com/single-cell-gene-expression/software/pipelines/latest/installation |
| Seurat v4.0 | Hao and Hao et al, 2021 | https://satijalab.org/seurat/articles/integration_introduction |
| ArchR package v1.0 | Granja et al, 2021 | https://www.nature.com/articles/s41588-021-00790-6#citeas |
| SCTransform | Hafemeister and Satija, 2019 | https://doi.org/10.1186/s13059-019-1874-1 |
| MACS2 V2.1.1 | Zhang et al, 2008 | https://doi.org/10.1186/gb-2008-9-9-r137 |

| Reagent/Resource | Reference or Source | Identifier or Catalog Number |
|---|---|---|
| Monocle R package v2.22.0 | Trapnell et al, 2014 | https://doi.org/10.1038/nbt.2859 |
| SCENIC package v1.3.1 | Aibar et al, 2017 | https://doi.org/10.1038/nmeth.4463 |
| **Other** | | |
| Chromium Single Cell 3′ Library & Gel Bead Kit v2, 16 rxns | 10X Genomics | Cat#120237 |
| Chromium Single Cell A Chip Kit, 48 rxns | 10X Genomics | Cat#120236 |
| Chromium i7 Multiplex Kit, 96 rxns | 10X Genomics | Cat#120262 |
| RNeasy Mini Kit | QIAGEN | Cat#741404 |
| TaqMan Fast Universal PCR Master Mix (2X), no AmpErase UNG | Thermo Fisher Scientific | Cat#4364103 |
| High-Capacity cDNA Reverse Transcription Kit | Applied Biosystems | Cat#4368814 |
| DAPI | Sigma-Aldrich | Cat#D9542 |
| Transwells | Corning | Cat#3470 |
| Qubit dsDNA HS Assay Kit | Invitrogen | Cat#Q32851 |
| GlutaMAX (100x) | Thermo Fisher Scientific | Cat#35050-061 |

## Mice

8–12-week-old Kras$^{LSL-G12D/WT}$; P53$^{flox/flox}$; Rosa26$^{LSL-eYFP}$ (KPY) mice were used to establish tumor organoids. Around 8–10-week-old athymic nude mice were used for transplantation assay. All mice were maintained in virus-free conditions and all mouse works were approved by the BCH Animal Care and Use Committee, accredited by AAALAC, and performed in accordance with relevant institutional and national guidelines and regulations.

## Isolating AT2 cells from mouse lung

Mouse lungs were dissected and digested for isolation of alveolar type 2 cells as previously described (Lee et al, 2014). Briefly, mice were anesthetized with Avertin and fixed on the dissection platform. Expose mouse lungs and hearts. Perfuse the mouse lung with 10–15 ml PBS (ice cold) and intratracheal inject 2 mL dispase (Corning). Cut the lung out and minced into small pieces. The lung pieces were digested with 0.0025% DNase (Sigma-Aldrich) and 100 mg/mL collagenase/dispase (Roche) in PBS for 45 min at 37 °C. The cell mixture was gently vortexed 2-3 times in the meantime. The cell mixture was filtered sequentially with 100 μm and 40 μm cell strainers (Falcon). The cell mixture was centrifuged at 1000 rpm for 5 min at 4 °C. The supernatant was discarded and the cell pellet was resuspended with red blood cell lysis buffer (0.15 M NH4Cl, 10 mM KHCO₃, 0.1 mM EDTA) for 90 s at room temperature. The lysis was quenched with a 30 ml PF10 buffer (10% FBS in PBS). The cell mixture was centrifuged at 1000 rpm for 5 min at 4 °C. The cell pellet was resuspended with 1 ml PF10 buffer. Four FACS antibodies, including anti-CD31-APC (Biolegend, 551262), anti-CD45-APC (Biolegend, 559864), anti-Ly-6A/E (SCA1)-APC/Cy7 (Biolegend, 560654), anti-CD326 (EpCAM) PE/Cy7 (Biolegend, 118216), were used at 1:100 concentration. DAPI (Sigma-Aldrich, 1:100) was used to eliminate dead cells. Single staining controls and fluorescence minus one (FMO) controls for all four channels were used to help decide the gate. FACS experiments were run on FACSAria II and FACS data were analyzed on FlowJo (BD).

## Induce AT2 cells and growing tumor organoids

To induce AT2 cells isolated from KPY mice, freshly sorted cells were counted and resuspended at 1 million cells/ml concentration in MTEC/Plus media (Zhang et al, 2017) containing $6 \times 10^7$ PFU/ml of Ad5CMV-Cre. The cell and virus mixture were incubated at 37 °C with 5% CO₂ for 1 h. Then wash the virus out with ice-cold PBS. Cells were resuspended in 3D media (Dulbecco's Modified Eagle's Medium/F12 (Invitrogen) supplemented with 10% FBS, penicillin/streptomycin, 1 mM HEPES, and insulin/transferrin/selenium (Corning)) at a concentration to be 100,000 cells/1 ml. Neonatal lung mesenchymal cells were used as supporting cells as previously described (Lee et al, 2014). The mesenchymal cells were suspended in growth factor reduced (GFR) Matrigel at a concentration of 1,000,000 cells/1 ml. The same volume of cell/3D media mixture and mesenchymal cell/Matrigel mixture was mixed and pipetted several times before adding 100 μl mixture into a Transwell (Corning). The mixture was solidified for 20 min at 37 °C, 5% CO₂. Add 500 μl 3D media into the bottom of the transwell. The media was changed every alternate day.

## Dissociation of tumor organoids for FACS

The 3D media was removed from the bottom of the transwell and the bottom well was washed with 500 μl PBS once. About 100 μl warm dispase (Corning) was added on the top of the transwells. The plate was kept at 37 °C, 5% CO₂ for 1 h. Next, 200 μl pipette tips were used to disrupt the matrigel structure and move all organoids mixture into a 1.5 ml falcon tube. The organoids were pelleted with a short spin and resuspended in TrypLE buffer for 5–10 min. The cells are monitored every 3 min under a microscope to make sure that a single-cell suspension forms. The lysis prep was quenched with PF10 buffer. Anti-CD44-APC (Thermo Scientific, 17-0441-82) was used to subset different cell states in the tumor organoids. Single staining control and FMO controls were used for the FACS experiment.

## Single-cell multi-omic sequencing

After digesting tumor organoids into single cells as described in "Dissociation of tumor organoids for FACS", scMulti-omic sequencing was performed using the 10X genomics platform (Chromium Next GEM Single Cell Multiome ATAC + Gene Expression kit, PN-1000285). Briefly, single cells from tumor organoids were used for nuclei isolation. Then the nuclei suspension was incubated with Transposition mix (10X Genomics) for DNA fragmentation in the open chromatin region. Then the transposed Nuclei were encapsulated on a 10X Genomics Chromium Controller with Chromium Next GEM Chip J. After pre-amplification, the RNA library and ATAC library were prepared separately. Quality control of RNA and ATAC libraries were run on the Agilent TapeStation High Sensitivity D5000 ScreenTape System, performed by Biopolymers Facility at Harvard Medical School. Libraries were sequenced by Bauer Core Facility of Harvard University using NovaSeq 6000 System.

 

## RNA extraction and quantitative RT-PCR

CD44-Neg and CD44-High population were sorted from 7-day tumor organoids as described in "Dissociation of tumor organoids for FACS". Absolutely RNA Microprep Kit (Agilent) was used to extract RNA. Then complementary DNA (cDNA) was synthesized using the SuperScript III Kit (Invitrogen). RT-PCR was performed with TaqMan Assays and software as per the manufacturer's recommendations.

## Recombination PCR

KPY or KPT organoids were grown as previously described, dissociated into single cells at day 7 in culture, and FACS sorted for DAPI-/EpCAM+/YFP+ or DAPI-/EpCAM+/Tdtomato+ and additionally for CD44-low and CD44-high. 100,000 cells were sorted for each condition. Cells were lysed and genomic DNA was extracted using the Zymo Quick-DNA™ MicroPrep kit (Cat. No. D3021, Lot No. 217088). 1 uL of gDNA was used in PCR reactions to amplify WT and mutant Kras and p53 alleles, with Promega GoTaq Green PCR mastermix (Cat. No. M7122).

Kras primers are as follows: Kras-primer-1 GTCTTTCCCCAG-CACAGTGC; Kras-primer-2 CTCTTGCCTACGCCACCAGCTC; Kras-primer-3 AGCTAGCCACCATGGCTTGAGTAAGTCTGCA. The expected band sizes for the Kras PCR reaction are as follows: WT Kras allele is 622 bp; unrecombined Kras-LSL-G12D allele is 500 bp; Cre-recombined Kras-LSL-G12D allele is 650 bp.

P53 primers were as follows: P53-primer-A CACAAAAA-CAGGTTAAACCCAG; P53-primer-B AGCACATAGGAGGCA-GAGAC; P53-primer-D GAAGACAGAAAAGGGGAGGG. The expected band sizes for the p53 PCR reaction are as follows: WT p53 allele is 288 bp; unrecombined p53-floxed allele is 370 bp; Cre-recombined p53-floxed allele is 612 bp.

## Orthotopic transplantation assay of tumor organoids

Around 8–12-week-old nude mice were used for transplantation assay. 1.5 U/kg bleomycin was injected intratracheally 24 h before the transplantation. CD44-Low and CD44-High populations were sorted from 7-day KPY tumor organoids as described in "Dissociation of tumor organoids for FACS". 50 K CD44-Neg or CD44-High cells suspended in 45 μl PBS were intratracheally administered into the bleomycin-injured mouse lung. Recipient mice were sacrificed 2 months later after transplantation for histology evaluation.

## Organoid-forming assay in co-culture and mono-culture condition

Primary KPY organoids (Passage 0) were digested into single cells and subsetted based on CD44 expression using FACS as described in "Dissociation of tumor organoids for FACS". Then, CD44-High and CD44-Neg populations were plated again and grown into organoids in co-culture or mono-culture conditions. For Co-cultured conditions, 2 K CD44-Low or CD44-High cells were suspended in 50 μl 3D media and mixed with 50 μl Matrigel containing 50k lung mesenchymal cells, then plated in one transwell. For the mono-culture condition, 2 K CD44-Low or CD44-High cells were plated in 50 μl 3D media and mixed with 50 μl Matrigel, then plated in one transwell. Around 500 μl 3D media was added to the bottom of the transwells at both conditions and changed every alternate day.

## Immunofluorescence and H&E

For tumor organoid staining, transwells were fixed with 10% neutral-buffered formalin overnight at room temperature, followed by dehydration by 70% ethanol overnight at room temperature. Carefully, the Matrigel plugs with organoids were moved out of the transwell inset and immobilized with Histogel (Thermo Scientific). These were sent for paraffin embedding and cut into 5 μm sections. Sections were rinsed in 100%, 95%, and 70% xylene successively for deparaffinization, and then rehydrated with 100%, 95%, and 70% ethanol. Next, sections were processed for hematoxylin and eosin (HE) staining or IF staining. H&E stainings were analyzed by at least two investigators, including a pathologist with expertise in murine lung cancer (Curtis SJ et al, Cell Stem Cell. 2010; Rowbotham et al, 2018). For IF staining, antigen retrieval was performed by incubating organoid sections in citric acid buffer at 95 °C for 20 min. The sections were washed with PBS-T (0.2% Triton X-100 diluted in PBS) three times and then blocked with Block buffer (10% donkey serum in PBS-T) for 1 h at room temperature. Several primary antibodies were used for IF staining, including CD44 (Thermo Scientific, 14-0441-82, 1:500), YFP (Abcam, ab6673, 1:400), SPC (Abcam, ab211326, 1:1000), Nkx2-1 (Abcam, ab76013, 1:250), Hmga2 (GeneTex, GTX629478, 1:200). The primary antibodies were diluted in blocking buffer and then incubated with section overnight at 4 °C. The sections were washed three times with PBS-T and incubated with secondary antibodies for 2 h at room temperature. All secondary antibodies are from Invitrogen and used at 1:200 dilution, including anti-Rabbit Alexa 488, anti-Rabbit Alexa 594, anti-Mouse Alexa 594, anti-rat Alexa 647, anti-goat Alexa 488. Mount the slides with Prolong Gold with DAPI (Invitrogen)

## Cytospin assay

For cytospin assay, organoids were digested into single cells and subsetted based on CD44 expression using FACS as described in "Dissociation of tumor organoids for FACS". Then, CD44-High and CD44-Neg populations were subjected to cytospin for 5 min at 300 rpm. Each cell area on the slides were washed with PBS-T (0.2% Triton X-100 diluted in PBS) three times and then blocked with Block buffer (10% donkey serum in PBS-T) for 1 h at room temperature. Several primary antibodies were used for IF staining, including P-Stat3 (Cell Siganling Technology, 9145T, 1:100), Stat3 (Cell Siganling Technology, 9139T, 1:800). The primary antibodies were diluted in blocking buffer and then incubated with the cells on the glass slides overnight at 4 °C. The slides were washed three times with PBS-T and incubated with secondary antibodies for 2 h at room temperature. All secondary antibodies are from Invitrogen and used at 1:200 dilution, including anti-Rabbit Alexa 488, anti-Rabbit Alexa 594, anti-Mouse Alexa 594, anti-rat Alexa 647, anti-goat Alexa 488. Mount the slides with Prolong Gold with DAPI (Invitrogen).

## Single-cell multiome data processing and dimension reduction

Cellranger-arc (v 2.0.1) mkfastq was used to demultiplexes raw base call data to fastq files, and then cellranger-arc count program was used to map single-cell RNA and single-cell ATAC reads to the mm10 mouse reference genome. The cell-by-gene expression

matrix was generated for the scRNA-seq data and the fragment files were generated for the scATAC-seq data as the outputs.

The scRNA-seq matrix was loaded into Seurat (Hao et al, 2021) (v4.0) package in R for further quality control and processing. We applied the following filtrations at the per-cell level to keep the high-quality cells: number of genes > 200 & number of reads > 2500 & percent of mitochondrial reads <10%. Cell cycle scores were regressed out during data scaling following the vignette of Seurat. For clustering, we took the first 2000 variable genes and performed PCA dimension reduction, followed by graph-based clustering and UMAP visualization using the top 60 principal components.

For the scATAC analysis, we utilized the ArchR package (Granja et al, 2021) (v1.0) to read in the fragment files, and then performed quality controls by requiring the minimal number of fragments to be 1000 and the minimal TSS enrichment to be 4. Dimension reduction was carried out using iterative LSI (Latent Semantic Indexing) with the 500 bp tile matrix generated by ArchR. The parameters of iterative LSI were set as follows: iterations = 3, resolution = 0.2 and varFeatures = 50,000. The ATAC cells were visualized using UMAP with the first 30 PCs from iterative LSI. Cell states annotation was based on both scRNA and scATAC signature as described in the main text.

### Integration of scRNA-seq data

Integration was carried out between the RNA data of KPY scMulti-omic profiling and the scRNA data of KY profiling. We firstly used SCTransform to perform count matrix normalization and identify the most variable genes, with the percentage of mitochondria regressed out. We then fed in the top 3000 most variable genes as anchors to perform integration between the two scRNA datasets. UMAP was used to visualize the integrated single-cell map.

### Differentially expressed genes and differential accessible peaks

Differentially expressed genes (DEGs) were identified with Wilcoxon rank-sum test using the FindMarkers function of Seurat. The minimal percent of expression was set to 0.05, and the log fold change threshold was set to 0.5. Peak calling was carried out using MACS2 (v 2.1.1) on the pseudobulk of each cell state, with the adjusted $p$-value threshold set as 0.01. Differential accessible peaks (DAPs) were identified with Wilcoxon test using the getMarkerFeatures function of ArchR, with TSS enrichment and number of fragments considered as confounding factors.

### Transcription factor analysis

Transcription factor (TF) analysis was performed with three different approaches, including single-cell chromVar-based enrichment, scenic-based regulon, and cell-state-specific TF expression. We first performed chromVar-based motif deviation analysis using the computeDeviations function of ArchR, with the genome-wide TF motif binding annotation from the cisbp database. We then used SCENIC package (v 1.3.1) to establish the TF-centric regulatory network, with cisTarget database used as TF target annotation. Lastly, we examined the expression of the TFs identified from the above two approaches. The TFs that are consistent across all three axes of motif enrichment, regulatory network and gene expression were defined as the TF regulators in each cell state.

### Trajectory analysis

We used the Monocle R package (v 2.22.0) to construct the trajectory of single cells at the per-cell level. The genes expressed in less than 10% of total cells were removed. The top 2000 differentially expressed genes between the SPC-high vs Hmga2-high were used to construct the pseudotime trajectory. The cell-cycle-related genes were removed from the differentially expressed gene list to minimize the effect from the cell cycle when constructing the trajectory. In addition, we used the Velocyto (v 0.17.17) to construct RNA velocity trajectory and expression dynamics, and then used the SeuratWrappers R package for visualization.

### Statistical analysis

For the boxplots showing the expression of genes, the boxes show the 25th–75th percentile, the lines show the median value, and the whiskers show the 1.5x interquartile range (IQR). For the bar plots with error bars, the data are demonstrated as mean ± SEM. False discovery rate (FDR) is applied for multi-testing correction for the GO functional annotation analysis. Asterisks in the figures denote the significance of $P$-values: *$p$ value < 0.05; **$p$ value < 0.01; ***$p$ value < 0.001; n.s. $p$ value > 0.05. R language was utilized for making the figures and performing the statistical analysis.

## Data availability

The raw single-cell RNA and single-cell ATAC matrices are available at Zenodo: https://doi.org/10.5281/zenodo.7713052. Code for single-cell multi-omic Seq analysis available at: https://github.com/lijingyun-zju/scMultiome_KPY_2022.git. Both the data and code are freely available with no restriction.

The source data of this paper are collected in the following database record: biostudies:S-SCDT-10_1038-S44318-025-00376-6.

## Peer review information

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

## Acknowledgements

We thank members from Kim lab for discussion and feedback. We thank the flow cytometry core facility at Boston Children's Hospital (BCH), the single-cell core facility at Harvard Medical School (HMS), the HMS biopolymers facility, and the DFHCC rodent histopathology facility. Graphical abstract was created with BioRender.com. SS is the Deborah J. Coleman Fellow of the Damon Runyon Cancer Research Foundation (DRG-2523-24). This work was supported in part by a Boehringer Ingelheim Fonds PhD fellowship (AFMD), the Damon Runyon Cancer Research Foundation postdoctoral fellowship (no. DRG:2368-19) and a Postdoctoral Enrichment Program Award from the Burroughs Wellcome Fund (no. 1019903) (ALM), F31 HL159919 (IGW), Landry Cancer Consortium fellowship (SMD), LUNGevity Foundation HEI Research Fellow Award 2023-10 (MFT), and the American Cancer Society Research Scholar Grant RSG-08-082-01-MGO, the V Foundation for Cancer Research, the Thoracic Foundation, the Ellison Foundation, the American Lung Association LCD-619492 R35HL150876, P50CA265826, R01CA233671, The G. Harold & Leila Y. Mathers Foundation, a gift from Winn-Barlow, and the Harvard Stem Cell Institute (CFK).

## Author contributions

**Jingyun Li**: Data curation; Software; Formal analysis; Investigation; Visualization; Methodology; Writing—original draft; Project administration. **Susanna M Dang**: Validation; Investigation; Writing—review and editing. **Shreoshi Sengupta**: Validation; Investigation; Writing—review and editing. **Paul Schurmann**: Validation. **Antonella F M Dost**: Investigation; Writing—review and editing. **Aaron L Moye**: Methodology; Writing—review and editing. **Maria F Trovero**: Funding acquisition. **Sidrah Ahmed**: Investigation. **Margherita Paschini**: Writing—review and editing. **Preetida J Bhetariya**: Software; Writing—review and editing. **Roderick Bronson**: Methodology. **Shannan J Ho Sui**: Writing—review and editing. **Carla F Kim**: Conceptualization; Resources; Supervision; Funding acquisition; Project administration; Writing—review and editing.

Source data underlying figure panels in this paper may have individual authorship assigned. Where available, figure panel/source data authorship is listed in the following database record: biostudies:S-SCDT-10_1038-S44318-025-00376-6.

## Disclosure and competing interests statement

We declare no competing interests. CFK and ALM are founders of Cellforma. CFK is a member of the *EMBO Journal* editorial advisory board.

# Expanded View Figures

**Figure EV1.**   related to Fig. **1**. **Cell state definition by combining gene expression assay and chromatin accessibility assay from one single cell.**

(**A**) UMAP projection of scMulti-omic gene expression data of KPY and YFP control organoids. Cells are colored by RNA clusters. (**B**) UMAP projection of scMulti-omic gene expression data of KPY and YFP control organoids. Cells are colored by sample ID. (**C–E**) Umap plots highlighting expression level (Log(TPX + 1), color bar of Umaps and Y axis of boxplots) of selected genes. (**F**) UMAP projection of scMulti-omic gene expression data of KPY organoids. Cells are colored by RNA clusters. (**G**) UMAP projection of scMulti-omic chromatin accessibility data of KPY organoids. Cells are colored by cell clusters identified in Fig. EV1F. (**H**) Each cluster identified from scMulti-omic gene expression data (Fig. EV1F) is illustrated in Chromatin accessibility UMAP. (**I**) Heatmap showing the highly expressed genes in each group of cells. (**J**) Umap plots highlighting expression level (Log(TPX + 1), color bar of Umaps and Y axis of boxplots) of selected genes.

                                    

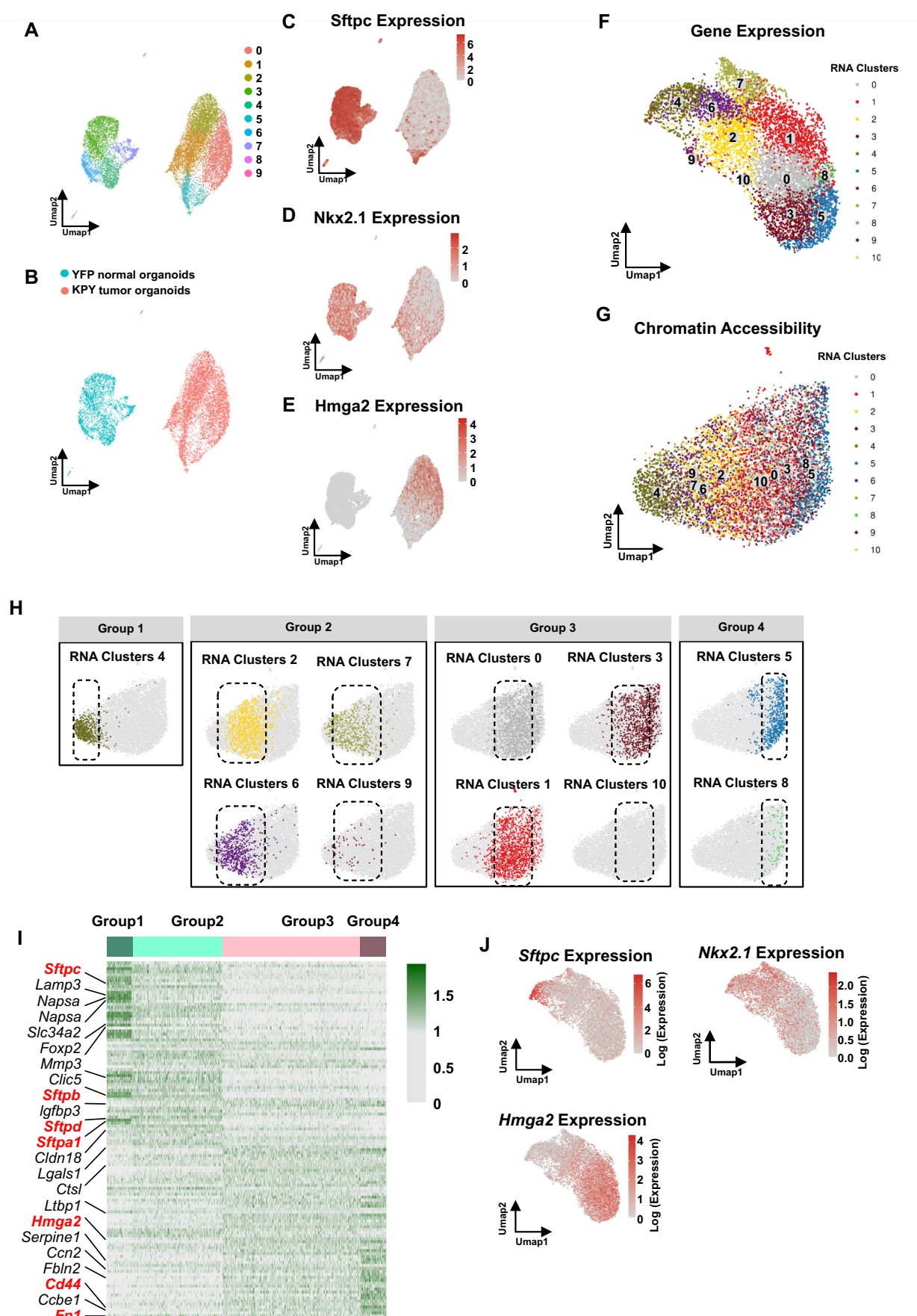

**A** *Nkx2-1* Gene Expression

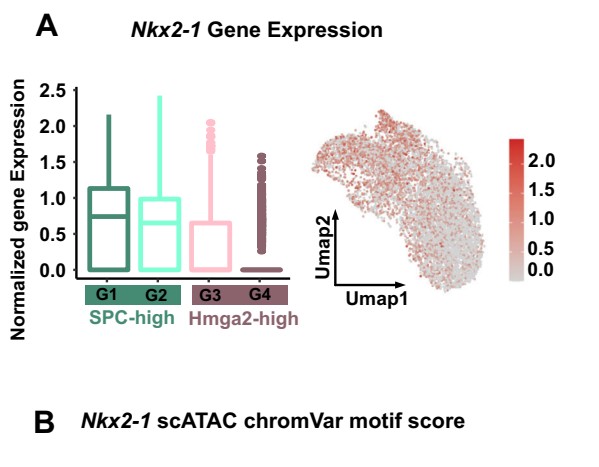

**D** Nfkb1 Gene Expression

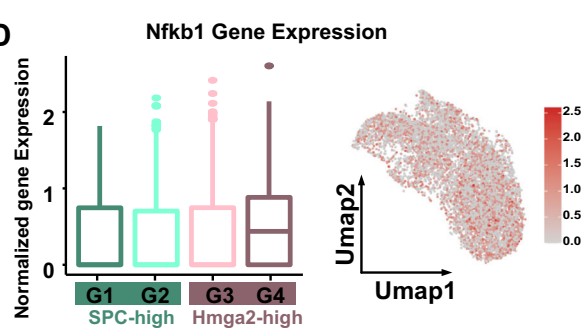

**B** *Nkx2-1* scATAC chromVar motif score

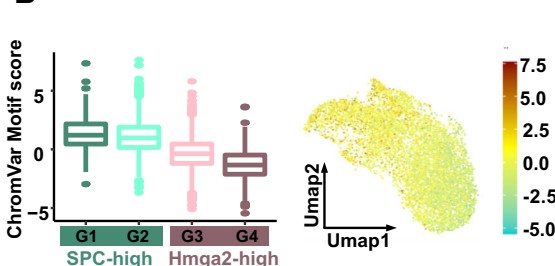

**E** Nfkb1 scATAC chromVar motif score

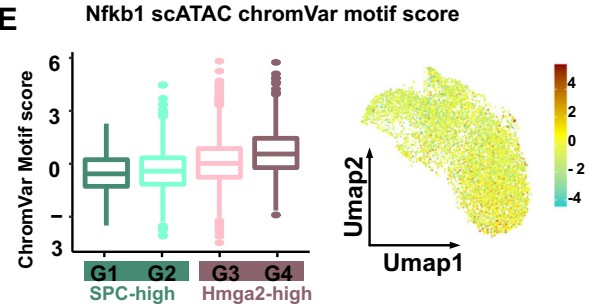

**C** *Nkx2-1* Regulon Expression

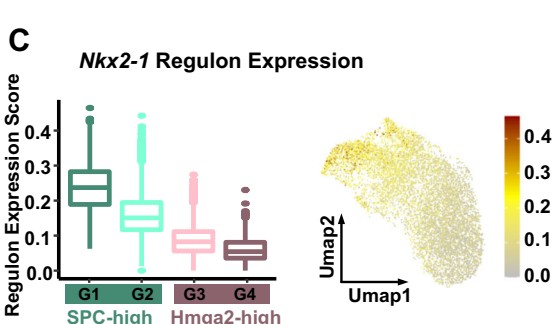

**F** Nfkb1 Regulon Expression

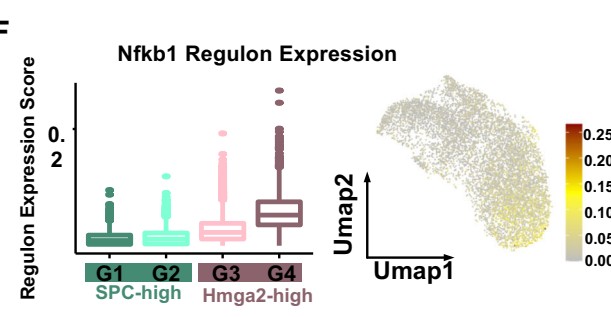

**G** scATAC TFs chromVar motif score

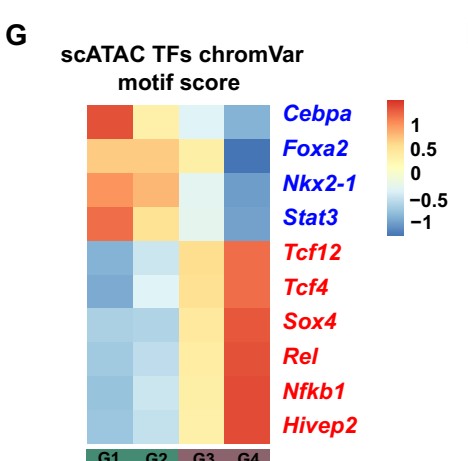

**H** TFs Regulon Expression

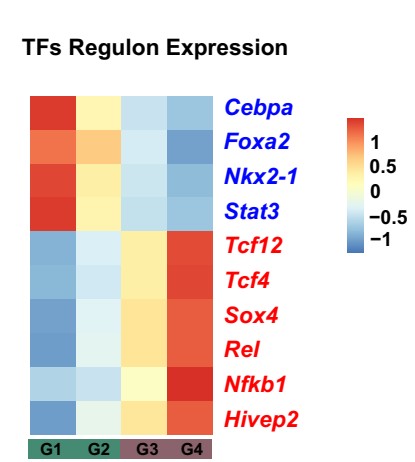

**I**

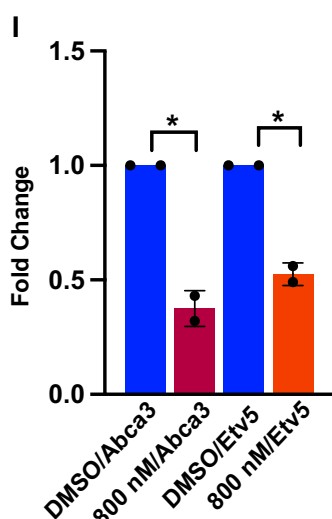

◀

**Figure EV2.** related to Fig. **2**.

(**A–C**) Showing candidate regulator for SPC-high cells. The gene expression level (**A**), motif enrichment score (**B**) and Regulon expression score (**C**) of Nkx2.1 are shown both in boxplots and Umap. The central line represents the median, the box encompasses the interquartile range (IQR) (25th to 75th percentile), and the whiskers extend to the minimum and maximum values within 1.5 times the IQR. Outliers are shown as individual points beyond the whiskers. G1: $n = 773$; G2: $n = 2514$; G3: $n = 4367$; G4: $n = 769$. (**D–F**) Showing candidate regulator for Hmga2-high cells. The gene expression level (**D**), motif enrichment score (**E**), and regulon expression score (**F**) of Nfkb1 are shown both in boxplots and Umap. The central line represents the median, the box encompasses the interquartile range (IQR) (25th to 75th percentile), and the whiskers extend to the minimum and maximum values within 1.5 times the IQR. Outliers are shown as individual points beyond the whiskers. G1: $n = 773$; G2: $n = 2514$; G3: $n = 4367$; G4: $n = 769$. (**G, H**) Summary of all candidate regulators for SPC-high and Hmga2-high cells. The motif enrichment score (**G**) and regulon expression score (**H**) for each candidate regulator are shown using heatmaps. (**I**) qPCR showing that Stattic treatment (800 nM) on day 7 KPY organoid cells reduced the expression of putative Stat3 target genes Abca3 and Etv5, compared to DMSO treatment. The data represents the mean ± SD ($n = 2$). P-value was calculated using an unpaired t-test with Welch's correction. *p*-Values = 0.035 and 0.046 (from left to right).

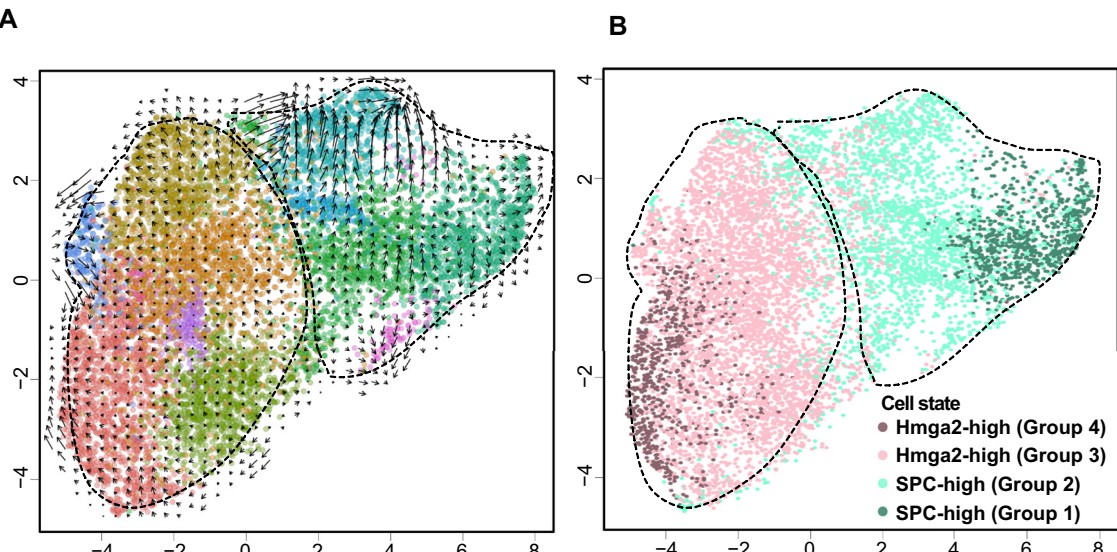

**Figure EV3.    related to Fig. 3. Pseudotime analysis reconstructs tumorigenesis trajectory in tumor organoids.**

(**A**) RNA velocity analysis of 7 day KPY tumor organoids. (**B**) Cell states identity are plotted on the RNA velocity UMAP.

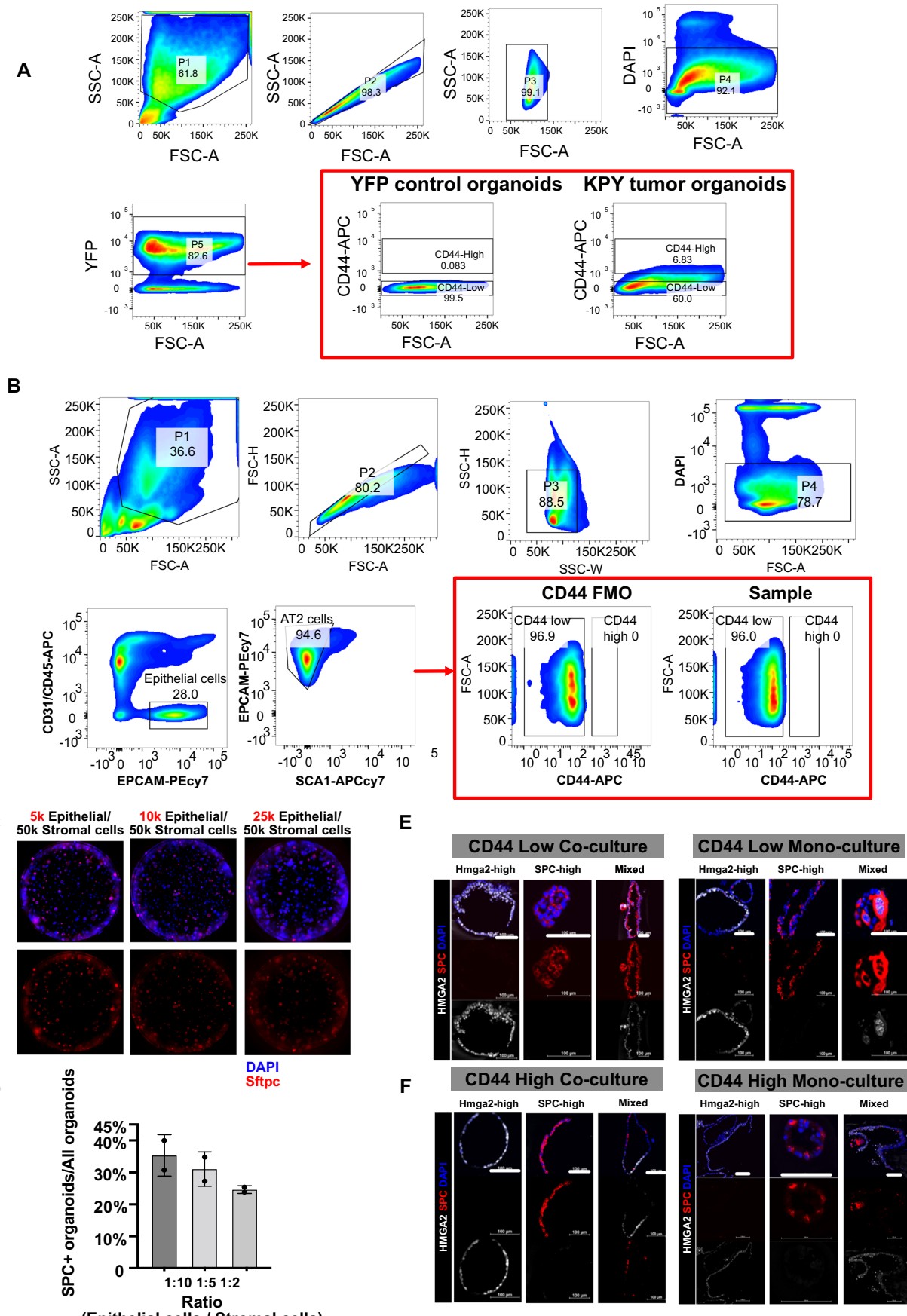

◀ **Figure EV4.** related to Fig. 4. **Co-culture with lung mesenchymal cells enhanced the organoids forming ability of SPC-high cells but not Hmga2-high cells.**

(A) FACS strategy for subsetting two cell states from 7 days KPY organoids using CD44. (B) Check the expression of CD44 in freshly sorted AT2 cells (DAPI-/CD31-/CD45-/EPCAM+/SCA1-). CD44 FMO control was used to set the CD44-neg and CD44-high gate. (C) Representative pictures of whole mount staining on 7 days KPY organoids in different conditions when the ratio between epithelial cells and mesenchymal cells is 1:10, 1:5, 1:2. (D) Bar plot showing the percentage of SPC+ organoids in three conditions when the ratio between epithelial cells and mesenchymal cells is 1:10, 1:5, 1:2. The data represents the mean ± SD. (E) Representative pictures of 7 days SPC+/Hmga2- organoids (SPC-high), SPC-/Hmga2+ organoids (Hmga2-high) and SPC + /Hmga2+ organoids (Mixed) derived from CD44-neg population in Co-culture and Mono-culture condition. Scale bar, 100 μm. (F) Representative pictures of 7 days SPC+/Hmga2- organoids (SPC-high), SPC-/Hmga2+ organoids (Hmga2-high) and SPC+/Hmga2+ organoids (Mixed) derived from CD44-high population in Co-culture and Mono-culture condition. Scale bar, 100 μm.

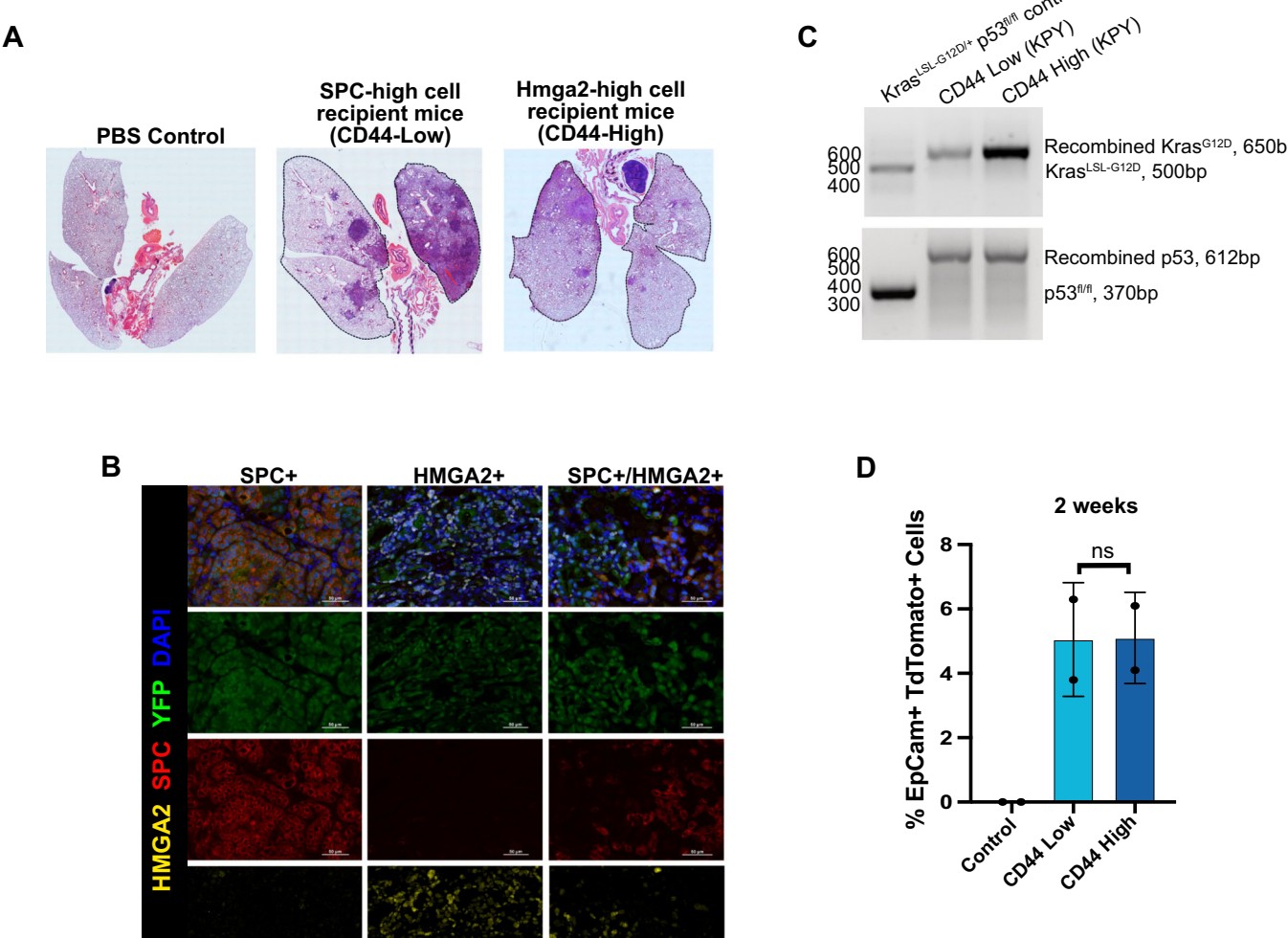

**Figure EV5.   related to Fig. 5. SPC-high cells have higher tumorigenic capacity than Hmga2-high cells in vivo.**

(A) HE staining of lungs from PBS control, CD44-neg, and CD44-high recipient mice. (B) IF staining showing the expression of SPC and Hmga2 in lesions from tumor organoids recipient mice. Both CD44-neg recipient mice and CD44-high recipient mice can derive SPC+, HMGA2+, SPC+/HMGA2+ tumors. Scaled bar = 50 μM. (C) Recombination PCR data showing no amplified bands corresponding to the unrecombined Kras or p53 alleles in both the CD44-low and CD44-high KPY cell population. (D) Bar diagram quantifying the percentage of DAPI-/EpCam+/TdTomato+ from mice injected with CD44-low and -high organoid cells. N.s. non-significant. The data represents the mean ± SD. *p*-value was calculated using an unpaired t-test with Welch's correction. *p*-Value = 0.9780 (compare light blue vs. dark blue bars).

