## [Peer Review File · The EMBO Journal]

Organoid modeling reveals the tumorigenic potential of the alveolar progenitor cell state

Jingyun Li, Susanna Dang, Shreoshi Sengupta, Paul Schurmann, Antonella Dost, Aaron Moye, Maria Trovero, Sidrah Ahmed, Margherita Paschini, Preetida Bhetaria, Roderick Bronson, Shannan Ho Sui and Carla Kim

Corresponding authors: Carla Kim (carla.kim@childrens.harvard.edu) , Jingyun Li (lijingyun@zju.edu.cn)

Review Timeline:

Submission Date:	29th Feb 24
Editorial Decision:	1st Mar 24
Revision Received:	17th Oct 24
Editorial Decision:	3rd Dec 24
Revision Received:	16th Jan 25
Accepted:	23rd Jan 25

Editors: Kelly Anderson / Daniel Klimmeck

Transaction Report:

(The initial review process for this manuscript took place with another journal. The initial reviewers' comments and authors' responses for this article have been made available. With the exception of the correction of typographical or spelling errors that could be a source of ambiguity, letters and reports are not edited. Depending on transfer agreements, referee reports obtained elsewhere may or may not be included in this compilation. Referee reports are anonymous unless the Referee chooses to sign their reports.)

Our point-by-point responses to the [other journal] reviewers' comments are as follows:

Overall Response: Thank you for the valuable feedback on our manuscript. We have considered the reviewers' concerns and performed additional experiments to improve our manuscript. In particular, we have worked to better develop the new biological insights towards understanding early-stage lung tumorigenesis that this manuscript provides.

The main conclusion of our work is that among the two cellular states we defined by multi-omic analysis of transcription and chromatin accessibility, the state that drives early-stage lung adenocarcinoma (LUAD) is that which most resembles the profile of the alveolar type II (AT2) cells. Many studies have revealed that during tumorigenesis, AT2 cells lose their differentiation state or even become more plastic; the dogma in the field is that advanced LUAD is characterized by an aggressive, EMT-like state that is less differentiated than AT2 cells. However, our work probes gene regulation at an earlier stage than previous work, examining the outcome of oncogenic Kras expression only seven days after activation in the normal progenitor cell (AT2 cells). This gave us the opportunity to learn that in early-stage LUAD, it is the AT2-like state that has more tumorigenic capacity compared to the EMT-like state. This is meaningful because the pathways that have been a focus of many LUAD therapeutic studies appear to be highly expressed in the EMT-like state, and understanding the molecules that regulate the AT2-like state may allow identification of more lung cancer targets.

Here we address the reviewers' specific comments:

Reviewer #1: (Comments for the author) The regulatory mechanism underlying the initiation of AT2 cells-derived lung cancer remains unclear. Li et al. employed single cell RNA-seq and ATAC-seq analyses to elucidate the changes of AT2 cells after the onset of Kras activation and P53 loss (KP) using previously established tumor organoid system. They identified two major cellular states in the KP tumor organoids, one resembling AT2-like (SPC-high) state and the other featured with loss of AT2 identity and the EMT-like (Hmga2-high) state. They further showed that the KP tumor organoids closely recapitulate the major cell states of the in vivo tumors at gene expression level as well as chromatin accessibility level. They found that the two cell states can be characterized by different transcription factor networks and represent as distinct path for tumorigenesis. Moreover, they identified CD44 as a marker for the Hmga2-high state and demonstrated that SPC-high cells had a higher tumorigenic capacity. This work is very interesting, well designed and technically sound. I list several concerns below for the authors to clarify.

1. The authors identified two major cell states, one with AT2-like (SPC-high) state and the other without AT2 identity (Hmga2-high). Interestingly, previous study has also identified two distinct subpopulations in KP model, one with high Wnt signalling activity and another forming a niche that provides the Wnt ligand (PMC5903678). I feel very curious if there is any correlation of these TWO different classifications.

Thank you for providing the valuable suggestion. As the reviewer has noted, Tammela et al (*Nature*, 2017) have defined distinct Wnt signaling populations in advanced LUAD in the same mouse model we used. We have examined the expression of key markers associated with Wnt-responsive cells, namely Lgr5, Ctnnb1, Lgr4, Wnt7a, Wnt7b, etc., as well as niche cells marked by Porcn (**Reviewer Figure 1 below**). Interestingly, we observed that Porcn exhibited elevated expression in the SPC-high Group1 cells, whereas Ctnnb1 and Wnt7b displayed higher expression in the other three populations, namely SPC-high Group2, Hmga2-high Group3, and Hmga2-high Group4. This finding suggests that SPC-high Group1, characterized by sustained expression of AT2 marker genes, might potentially represent Wnt niche cells. However, it is important to note that this interpretation would require substantial further investigation to test including the use of genetically engineered mice we do not have in our animal colony. We agree that this would be an interesting avenue for future studies.

Figure for reviewers removed

2. The author used the CD44-negative population to indicate the SPC-high cells. They found that CD44-negative (or SPC-high) cells had a higher organoid forming ability when co-cultured with lung stromal cells and had a higher tumorigenic capacity in vivo (Figure 4F-G, 5B-D and S5A). CD44 is commonly considered as the cancer stem cell marker and multiple studies have demonstrated that high CD44 expression significantly contributes to enhanced tumorigenic capacity in different cancer types. How do the authors reconcile their findings with previous observations?

While numerous studies have highlighted CD44 as a marker gene associated with cancer stem cells in a variety of cancer types, the majority of investigations that identify cancer stem cells primarily focus on later stages of tumorigenesis, whereas our study specifically examines cell states in the earliest stages of tumorigenesis. We have not examined CD44 as a marker of the cancer stem cell population (for which we prefer the term tumor-propagating cells) in advanced Kras p53 tumors. However, In 2010 we showed that Sca1, encoded by the gene *Ly6A*, is a surface marker that identifies tumor-propagating cells in the Kras LUAD model (Curtis et al, *Cell Stem Cell* 2010). We examined *Ly6a* in our data set for this manuscript and found that it is uniformly expressed in all groups of KPY cells at day 7 (not shown). Thus, it is possible that the cellular states that drive early-stage LUAD are different from the cells that can propagate the phenotype of advanced LUAD.

3. Some errors in the main text. For example, on line 202, the "Figure S2A-C" should be "Figure 2C"; on line

234, the "KP" should be "KPY"; on line 238, the "K" should be "KY".

Thanks for pointing this out. We have made changes accordingly.

Reviewer #2: In this manuscript, Jingyun Li et al., used organoid models and multiome profiling to study lung adenocarcinoma development in mice. Specifically, the authors cultured Kras/P53 mutant alveolar type-2 cells together with lung stromal cells in 3-dimensional organoid models and performed combined scRNA-seq and snATAC-seq on cells collected on day-7. The authors performed systematic analysis to identify transcription factors that are expressed in different cell types/states and correlated them to TF motif prediction and TF regulons (potential downstream targets). Using this data the authors identified 4 different clusters of cells, G1, G2, G3, and G4. Based the expression of Sftpc and Hmga2, the authors categorized these four clusters into two cell states. G1 and G2 were grouped into SPChi whereas G3 and G4 were grouped into Hmga2hi cells. Further, the authors claim that CD44 expression is highly enriched in Hmga2hi groups. The authors used CD44 to separate "SPChi cells" and "Hmga2hi cells". These sorted cells were then used for testing their ability to form tumor organoids in culture and tumors in in vivo transplantation model. Based on this data, the authors claim that SPChi cells shows significantly higher tumorigenic potential compared to Hmga2hi cells.

Overall, this manuscript does not provide much any new conceptual advances in tumorigenesis. Previous work including authors own papers have demonstrated the utility of organoids in studying tumorigenesis. Therefore, current manuscript does not provide technical or conceptual advances. The authors used multi-omic profiling to identify transcriptional programs in AT2s with constitutive active mutant Kras and TP53 deletion. However, similar profiling has been done by others in vivo. Therefore, this work does not add much. Further, there are some technical issues that may impact analysis and interpretation. See below for specific comments:

As mentioned by Reviewer 2, our previous work published in Dost et al. (*Cell Stem Cell* 2020) provides evidence for the efficacy of our epithelial-stromal co-culture organoid system in replicating crucial transcriptional changes in AT2 cells following expression of oncogenic *Kras*. This supports the utilization of organoid systems for studying the initial stages of tumor development. In that particular study, we observed that K tumor organoids contained cells that lost AT2 cell identity. However, in that study, we did not perform experiments to compare different cellular states in the K organoids. In this study, we explored the question in two layers, assessing gene expression and chromatin accessibility and testing the function of different cell states. We also used the Kras p53 model for this work. Here we demonstrate that our organoid system effectively reproduces the heterogeneity observed in in vivo KP tumors at both the transcriptional and chromatin accessibility levels. Moreover, we identified potential new key regulators associated with different states of tumor cells by combining RNA profiling and chromatin profiling. Significantly, our findings reveal that tumor cells retaining a more alveolar type II (AT2) cell identity subsequent to Kras activation and p53 loss exhibit a higher capacity for tumorigenesis compared to tumor cells that exhibit upregulation of epithelial-mesenchymal transition (EMT) programs. This finding is novel because it is the opposite of what is predicted—the prediction is that the cells that seem to be more akin to the EMT-like aggressive tumors cells would be more tumorigenic. Our findings underscore the significance of targeting AT2-like tumor cells (SPC-high tumor cells) as an avenue to impede tumor progression during the early stages of lung cancer. The pathways with higher expression and chromatin accessibility in the EMT-like cells are widely addressed in cancer research, yet the factors and pathways we highlight as being enriched in the AT2-like tumor cells are largely unexplored in the literature and thus represent novel potential therapeutic targets.

Major concerns:

1. The authors describe 4 different groups of cells based on expression of SFTPC, HMGA2, and other TF modules inferred from ATAC-seq data. However, it is not clear whether these subsets truly represent different

stages of tumorigenic events or is it due to different recombination events? For example, some cells may have only mutant Kras activation whereas others are Tp53 deletion only. Further, how does this pattern look at different time points? Combined single cell transcriptome and ATAC-seq data at different time points is necessary to better understand the dynamics of these cells. Data from controls such as KrasG12D, TP53, and WT cells are needed to directly compare and assess the differences between cells with different genetic contexts. This is essential to assess different recombination possibilities expected in these cultures.

We acknowledge the potential existence of distinct recombination events, even though we consider this probability to be minimal based on our data. The results presented in **Figures 3H and 3I** address these concerns. We observed all subsets in both KPY data sets as well as Kras-only tumor organoid cells, indicating that the identified subsets genuinely represent various stages of tumorigenesis rather than resulting from different recombination events that lead to differences in key genes such as p53.

To further address the reviewer's concern, we also asked whether there might be differences in recombination in the CD44-Neg/High fractions of KPY tumor organoid cells that could contribute to differences in tumorigenic capacity. KPY tumor organoids were dissociated, sorted based on CD44 levels, and used for DNA preparations. Recombination-specific PCR was performed, which showed that there was no detectable non-recombined Kras^{G12D} in either subset of organoid cells. Regarding p53, we did observe some instances of incomplete recombination, although this phenomenon appeared more in the CD44-Neg population (see Reviewer Figure 2). Since the CD44-Neg population was more tumorigenic in our studies, it is unlikely that any lack of total recombination, which would theoretically mean more p53 tumor suppressive activity, can be the reason for enhanced activity in the CD44-Neg cells.

Figure for reviewers removed

We also investigated whether freshly sorted AT2 cells (the source cells for deriving our organoids) exhibit heterogeneity in terms of CD44 expression. CD44 expression levels were assessed in freshly isolated AT2 cells using FACS (see Reviewer Figure 3). The results indicate that freshly sorted AT2 cells did not exhibit positivity for CD44.

Figure for reviewers removed

For the second concern about dynamic changes of the two subsets during tumorigenesis, we do have single cell RNA-seq dataset from a KPY tumor organoid time course, but those data are part of a separate manuscript (data not shown). In the referenced study, our investigation revealed the presence of the two subsets as early as 4 days, persisting until the last time point at 14 days, which represents the culmination of our data collection. Noteworthy is the discerned alteration in the ratio between these subsets throughout the progression of tumorigenesis. Furthermore, as the reviewer points out above, our findings correlate well with RNA and ATAC sequencing data from various time points in the published KP GEMM, supporting the idea that they are not simply a result of one snapshot in culture.

Thirdly, as the reviewer suggested, we agreed it was important to examine the omics data in WT cells as a comparison. We have now compared cells from KP tumor organoids and WT organoids using the multi-omics methods (Please see revised Figure S1A and S1B). The KP cells and WT cells distributed in separated clusters in UMAP, indicating dramatic transcriptional changes after oncogenic alteration. This also supports our conclusion that differences in clustering and cell states is not due to different levels of recombination.

2. Authors used Sftpc expression to classify Groups 1,2. However, data in Fig-1F shows Sftpc expression not much different in Group-2 compared to Group-3 and 4. Therefore, it's not clear why the authors grouped G1 and G2. Is G1 similar to unrecombined cells or WT cells? As mentioned in my comment #1, it is essential to directly compare current datasets with control cells.

Thanks for the nice suggestions. We grouped G1 and G2 for three reasons. Firstly, our differential gene analysis among the four groups of cells (Figure S1I) revealed a significant overlap in highly expressed genes between G1 and G2, indicating a notable similarity when compared to the other two groups. Notably, the highly expressed genes in G1 and G2 demonstrate enrichment in Gene Ontology (GO) terms associated with alveolar lamellar body and surfactant homeostasis (Figure 1D and E). This enrichment suggests that these

two groups of cells retain a portion of their original cellular function as alveolar type 2 cells, whose primary role in the normal lung is the synthesis and secretion of surfactant. Secondly, it is worth mentioning that both G1 and G2 still exhibit expression of the lung lineage regulator Nkx2.1, while G3 and G4 no longer express this gene. Numerous previous studies on KP lung adenocarcinoma provide substantial support for the notion that the loss of NKX2.1 expression is a crucial step in the progression of tumors (Figure 1F). Lastly, when comparing our in vitro organoids dataset with the in vivo KY tumor dataset, we observed that G1 and G2 were consistently grouped together at both the gene expression level and chromatin accessibility level (Figure 1I and 1J). Rather than naming the groups AT2-like or EMT-like, we referred to them by SPC and Hmga2 in order to keep nomenclature associated with gene expression of predominant markers that apply to the two major groups.

In an effort to enhance the robustness of our analysis, cells from control wild-type (WT) organoids have been incorporated. This inclusion has enabled us to affirm that G1 exhibits a greater similarity to recombined cells when contrasted with control cells, as delineated in the revised Figure S1A and S1B, presented above. While G1 demonstrates a heightened expression of Sftpc compared to G2, G3, and G4, the Sftpc expression level remains lower than that observed in control organoids.

3. Much of the data in this manuscript comes from correlation of gene expression and TF motif prediction. This manuscript does not provide any validation for these candidates using organoid models.

Thank you for the suggestion. Our integration analysis has revealed distinct transcription factor utilization in tumorigenesis between SPC-high and Hmga2-high cells. While many of the enriched transcription factors (TFs) have previously been rigorously validated for their significance in Kras mutant LUAD progression in vivo, our study highlights the cell state-specific roles of these TFs. For example, TFs highly expressed in the SPC-high cells and the Hmga2-high cells have been shown to be required for tumorigenesis in the Kras model, such as STAT3, Wnt, and NF κ B. The pro-tumor properties of Stat3 were further validated using our organoid system (refer to Reviewer Figure 4). Cyto-spin staining revealed that while CD44-negative and CD44-high cells exhibited comparable levels of Stat3 protein, the CD44-negative population displayed a higher percentage of phosphorylated Stat3 protein, indicating elevated Stat3 activity in the CD44-low population. Additionally, qPCR analysis indicated higher expression of Stat3 target genes in CD44-negative cells compared to CD44-high cells. Furthermore, we treated KP tumor organoids with a Stat3 inhibitor either on day 0 or day 3 after plating for co-culture. The results demonstrated that 5 μ M of the inhibitor significantly reduced organoid growth under both conditions. These findings confirm the role of Stat3 in promoting tumorigenesis.

We are currently conducting experiments to validate other candidate TFs and their cell state specificity both in vitro and in vivo. However, the comprehensive validation process necessitates a considerable amount of time, which extends beyond the scope of this manuscript.

Figure for reviewers removed

4. Tumor progression: Does organoid model recapitulate tumorigenesis in vivo? For example, do they progress through AHA, adenoma, and adenocarcinoma similar to in vivo?

Yes, the organoid system models in vivo tumorigenesis including progression in stages. The progression of tumorigenic phenotypes in our Kras and Kras/p53 tumor organoids has been described in our previous publication, Dost et al 2020. (Reviewer Figure 5 and 6). There we noted that “Histological analysis revealed that our tumor organoid model recapitulated in vivo tumor progression.” Notably, H&E analysis was done over

a time course. At days 7-10, Kras and Kras/p53 organoids demonstrated pleiomorphic nuclei similar to that observed in AHA and adenoma whereas wild-type organoid cells had normal nuclei. Giant multinucleated cells were observed at later time points in the tumor organoids, similar to those observed in KP adenocarcinomas in vivo at later stages of tumorigenesis.

In addition, upon transplantation via intratracheal administration in vivo, we observed the manifestation of various stages, including atypical hyperplasia (AHA), adenoma, and adenocarcinoma (Dost et al Figure 2(F–H) and this manuscript Figure 5C).

Figures for reviewers removed

5. In Fig3E-G, the authors show 3 types of organoids: SPChi, HMGA2hi, and mixed type. Are these differences due to cell of origin (i.e., subsets of AT2s) or different recombination events? Further, authors prior work has revealed that BASCs can generate mixed type (airway and alveolar) organoids. However, current study did not consider using different subsets of epithelial cells as starting population to assess any correlation between cell of origin and tumor organoid outcomes.

The concern regarding different recombination events has been addressed in response to reviewer question 1 above. Regarding the issue of a heterogeneous population with different subsets of AT2 cells being used

to develop the tumor organoids, we cannot rule out this possibility formally. However, our data suggest this is not the cause of the different types of organoids that emerge after oncogenic Kras expression. First, to address reviewer questions above, we have added data from multi-omics analysis of wild-type alveolar organoids. Notably, the cells in wild-type organoids cluster into one major cluster in contrast to the KPY organoid cells. Whereas some possible subclusters can be seen within the wildtype organoids, they are not distinguishable by SPC or Hmga2 as are tumor organoids (please see Figure S1), and they do not overlap with cells from KPY organoids.

Indeed, our laboratory has also characterized the BASC population as a cell of origin of LUAD. However, for this manuscript and our initial publication describing tumor organoids, we focused only on the use of AT2 cells to study the transcriptional changes that occur early after Kras is activated for the very reason that the reviewer points out. We were interested in studying cellular state, and we did not want to confuse the results with the variety of epithelial cell types that the BASC population can generate, which includes multipotent progenitors, AT2 cells and Club cells. It would be a very interesting to compare tumor organoids generated from BASCs and AT2 cells, yet will require substantial work beyond this manuscript scope. We added this to the Limitations section of our revised manuscript.

6. The finding that SPChi cells have higher tumorigenic potential than HMGAhi cells is interesting. However, its not clear what caused this? Is it due to differential engraftment of SPChi and HMGA2hi cells? Or is it due to positive or negative influence of support cells in the niche? Additionally, does the tumorigenic potential change in a different graft assay? For example, xenograft model or cells administered via tail vein.

We agree with the reviewer that SPChi cells have higher tumorigenic capacity than Hmga2hi cells is an interesting and unexpected finding and a highlight of our manuscript.

We are presently engaged in conducting supplementary in vivo transplantation experiments, wherein recipient mice are sacrificed at earlier time points to assess potential differences in engraftment ability between CD44-high and CD44-negative cells. We expect to obtain the results in the near future.

It is a very interesting question to ask if different subsets of KPY cells rely differently on supporting cells in the niche. Consistent with the in vivo transplantation result, our in vitro organoids forming assay suggests that SPChi cells exhibit a greater advantage in the presence of lung stromal cells (Figure 4F and 4G). In addition, an increased proportion of stromal cells in the co-culture environment correlates with a higher percentage of SPC+ organoids (Figure S4D). Given these results, we agree with the reviewer that the difference in tumorigenic potential between the two cellular states, at least in part, influenced by the supportive effects of lung niche cells. More detailed understanding of this aspect of our study will require more experiments beyond the scope of this manuscript.

The alternative graft assays suggested by the reviewer, such as subcutaneous or tail vein injection assays, are commonly employed to investigate topics related to tumor progression and metastasis, respectively, and indeed assess tumorigenesis in a distinct niche. Whereas these are very interesting studies that we are planning to conduct in the future, we wished to keep this manuscript focused on the earliest stages of tumorigenesis within the lung.

Dear Dr. Kim,

Thank you for submitting your manuscript for consideration by the EMBO Journal. We have now had a chance to read through your manuscript and point by point response and agree with the revision plan you have outlined. Therefore, I would like to invite you to submit a revised version of the manuscript.

Thank you for the opportunity to consider your work for publication. I look forward to your revision.

Yours sincerely,

Kelly M Anderson, PhD
Editor, The EMBO Journal
k.anderson@embojournal.org

We realize that it is difficult to revise to a specific deadline. In the interest of protecting the conceptual advance provided by the work, we recommend a revision within 3 months (30th May 2024). Please discuss the revision progress ahead of this time with the editor if you require more time to complete the revisions.

Response to reviewer's comments

Overall Response: Thank you for the valuable feedback on our manuscript. We have considered the reviewers' concerns and performed additional experiments to improve our manuscript. In particular, we have worked to better develop the new biological insights towards understanding early-stage lung tumorigenesis that this manuscript provides.

The main conclusion of our work is that among the two cellular states we defined by multi-omic analysis of transcription and chromatin accessibility, the state that drives early-stage lung adenocarcinoma (LUAD) is that which most resembles the profile of normal lung alveolar progenitors (alveolar type II or AT2 cells). Many studies have revealed that during tumorigenesis, AT2 cells lose their differentiation state or even become more plastic; the dogma in the field is that advanced LUAD is characterized by an aggressive, EMT-like state that is less differentiated than AT2 cells. However, our work probes gene regulation at an earlier stage than previous work, examining the outcome of oncogenic Kras expression only seven days after activation in the normal AT2 cells. This gave us the opportunity to learn that in early-stage LUAD, it is the AT2-like state that has more tumorigenic capacity compared to the EMT-like state. This is meaningful because the pathways that have been a focus of many LUAD therapeutic studies appear to be highly expressed in the EMT-like state, and understanding the molecules that regulate the AT2-like state may allow identification of new lung cancer targets especially for earlier intervention.

Here we address the reviewers' specific comments:

Reviewer #1: (Comments for the author) The regulatory mechanism underlying the initiation of AT2 cells-derived lung cancer remains unclear. Li et al. employed single cell RNA-seq and ATAC-seq analyses to elucidate the changes of AT2 cells after the onset of Kras activation and P53 loss (KP) using previously established tumor organoid system. They identified two major cellular states in the KP tumor organoids, one resembling AT2-like (SPC-high) state and the other featured with loss of AT2 identity and the EMT-like (Hmga2-high) state. They further showed that the KP tumor organoids closely recapitulate the major cell states of the in vivo tumors at gene expression level as well as chromatin accessibility level. They found that the two cell states can be characterized by different transcription factor networks and represent as distinct path for tumorigenesis. Moreover, they identified CD44 as a marker for the Hmga2-high state and demonstrated that SPC-high cells had a higher tumorigenic capacity. This work is very interesting, well designed and technically sound. I list several concerns below for the authors to clarify.

1. The authors identified two major cell states, one with AT2-like (SPC-high) state and the other without AT2 identity (Hmga2-high). Interestingly, previous study has also identified two distinct subpopulations in KP model, one with high Wnt signalling activity and another forming a niche that provides the Wnt ligand (PMC5903678). I feel very curious if there is any correlation of these TWO different classifications.

Thank you for providing the valuable suggestion. As the reviewer has noted, Tammela et al (*Nature*, 2017) have defined distinct Wnt signaling populations in advanced LUAD in the same mouse model we used. We have examined the expression of key markers associated with Wnt-responsive cells, namely Lgr5, Ctnnb1, Lgr4, Wnt7a, Wnt7b, etc., as well as niche cells marked by Porcn (**Reviewer Figure 1 below**). Interestingly, we observed that Porcn exhibited elevated expression in the SPC-high Group1 cells, whereas Ctnnb1 and Wnt7b displayed higher expression in the other three populations, namely SPC-high Group2, Hmga2-high Group3, and Hmga2-high Group4. This finding suggests that SPC-high Group1, characterized by sustained expression of AT2 marker genes, might potentially represent Wnt niche cells or their precursors. However, it is important to note that this interpretation would require substantial further investigation to test including the use of genetically engineered mice we do not have in our animal colony. We agree that this would be an interesting avenue for future studies.

Figure for reviewers removed

2. The author used the CD44-negative population to indicate the SPC-high cells. They found that CD44-negative (or SPC-high) cells had a higher organoid forming ability when co-cultured with lung stromal cells and had a higher tumorigenic capacity in vivo (Figure 4F-G, 5B-D and S5A). CD44 is commonly considered as the cancer stem cell marker and multiple studies have demonstrated that high CD44 expression significantly contributes to enhanced tumorigenic capacity in different cancer types. How do the authors reconcile their findings with previous observations?

While numerous studies have highlighted CD44 as a marker gene associated with cancer stem cells in a variety of cancer types, the majority of investigations that identify cancer stem cells primarily focus on later stages of tumorigenesis, whereas our study specifically examines cell states in the earliest stages of tumorigenesis. We have not examined CD44 as a marker of the cancer stem cell population (for which we prefer the term tumor-propagating cells) in advanced Kras p53 tumors. However, In 2010 we showed that Sca1, encoded by the gene *Ly6A*, is a surface marker that identifies tumor-propagating cells in the Kras LUAD model (Curtis et al, *Cell Stem Cell* 2010). We examined *Ly6a* in our data set for this manuscript and found that it is uniformly expressed in all groups of KPY cells at day 7 (not shown). Thus, it is possible that the cellular states that drive early-stage LUAD are different from the cells that can propagate the phenotype of advanced LUAD.

3. Some errors in the main text. For example, on line 202, the "Figure S2A-C" should be "Figure 2C"; on line 234, the "KP" should be "KPY"; on line 238, the "K" should be "KY".

Thanks for pointing this out. We have made changes accordingly.

Reviewer #2: In this manuscript, Jingyun Li et al., used organoid models and multiome profiling to study lung adenocarcinoma development in mice. Specifically, the authors cultured Kras/P53 mutant alveolar type-2 cells together with lung stromal cells in 3-dimensional organoid models and performed combined scRNA-seq and snATAC-seq on cells collected on day-7. The authors performed systematic analysis to

identify transcription factors that are expressed in different cell types/states and correlated them to TF motif prediction and TF regulons (potential downstream targets). Using this data the authors identified 4 different clusters of cells, G1, G2, G3, and G4. Based on the expression of Sftpc and Hmga2, the authors categorized these four clusters into two cell states. G1 and G2 were grouped into SPChi whereas G3 and G4 were grouped into Hmga2hi cells. Further, the authors claim that CD44 expression is highly enriched in Hmga2hi groups. The authors used CD44 to separate "SPChi cells" and "Hmga2hi cells". These sorted cells were then used for testing their ability to form tumor organoids in culture and tumors in an in vivo transplantation model. Based on this data, the authors claim that SPChi cells show significantly higher tumorigenic potential compared to Hmga2hi cells.

Overall, this manuscript does not provide much any new conceptual advances in tumorigenesis. Previous work including authors own papers have demonstrated the utility of organoids in studying tumorigenesis. Therefore, current manuscript does not provide technical or conceptual advances. The authors used multi-omic profiling to identify transcriptional programs in AT2s with constitutive active mutant Kras and TP53 deletion. However, similar profiling has been done by others in vivo. Therefore, this work does not add much. Further, there are some technical issues that may impact analysis and interpretation. See below for specific comments:

As mentioned by Reviewer 2, our previous work published in Dost et al. (*Cell Stem Cell* 2020) provides evidence for the efficacy of our epithelial-stromal co-culture organoid system in replicating crucial transcriptional changes in AT2 cells following expression of oncogenic Kras. This supports the utilization of organoid systems for studying the initial stages of tumor development. In that particular study, we observed that K tumor organoids contained cells that lost AT2 cell identity. However, in that study, we did not perform experiments to compare different cellular states in the K organoids, nor did we test the function of the cells with lost AT2 cell identity. In this study, we explored the question in two layers, assessing gene expression and chromatin accessibility and testing the function of different cell states. We also used the Kras p53 model for this work. Here we demonstrate that our organoid system effectively reproduces the heterogeneity observed in KP tumors in vivo at both the transcriptional and chromatin accessibility levels. Moreover, we identified potential new key regulators associated with different states of tumor cells by combining RNA profiling and chromatin profiling. Significantly, our findings reveal that tumor cells retaining a more AT2-like cell identity subsequent to Kras activation and p53 loss exhibit a higher capacity for tumorigenesis compared to tumor cells that exhibit upregulation of epithelial-mesenchymal transition (EMT) programs. This finding is novel because it is the opposite of what is predicted—the prediction is that the cells that seem to be more akin to the EMT-like aggressive tumor cells would be more tumorigenic. Our findings underscore the significance of targeting AT2-like tumor cells (SPC-high tumor cells) as an avenue to impede tumor progression during the early stages of lung cancer. The pathways with higher expression and chromatin accessibility in the EMT-like cells are widely addressed in cancer research, yet the factors and pathways we highlight as being enriched in the AT2-like tumor cells are largely unexplored in the literature and thus represent novel potential therapeutic targets.

Major concerns:

1. The authors describe 4 different groups of cells based on expression of SFTPC, HMGA2, and other TF modules inferred from ATAC-seq data. However, it is not clear whether these subsets truly represent different stages of tumorigenic events or is it due to different recombination events? For example, some cells may have only mutant Kras activation whereas others are TP53 deletion only. Further, how does this pattern look at different time points? Combined single cell transcriptome and ATAC-seq data at different time points is necessary to better understand the dynamics of these cells. Data from controls such as KrasG12D, TP53, and WT cells are needed to directly compare and assess the differences between cells with different genetic contexts. This is essential to assess different recombination possibilities expected in these cultures.

We acknowledge the potential existence of distinct recombination events, even though we consider this probability to be minimal based on our data. The results presented in **Figures 3H and 3I** address these concerns. We observed all subsets in both KPY data sets as well as Kras-only tumor organoid cells, indicating that the identified subsets genuinely represent various stages of tumorigenesis rather than resulting from different recombination events that lead to differences in key genes such as p53.

To further address the reviewer's concern, we also asked whether there might be differences in recombination efficiency in the CD44-Low/High fractions of KPY (or KPT) tumor organoid cells that could contribute to differences in tumorigenic capacity. KPY tumor organoids were dissociated, sorted based on

CD44 levels, and used for genomic DNA preparations. Recombination-specific PCR was performed, which showed that there was no detectable non-recombined Kras^{G12D} or non-recombined p53 in either subset of organoid cells (Manuscript Figure S5B and D).

We also investigated whether freshly sorted AT2 cells (the source cells for deriving our organoids) exhibit heterogeneity in terms of CD44 expression. CD44 expression levels were assessed in freshly isolated AT2 cells using FACS (see Reviewer Figure 2). The results indicate that freshly sorted AT2 cells did not exhibit positivity for CD44.

Figure for reviewers removed

For the second concern about dynamic changes of the two subsets during tumorigenesis, we do have a single-cell RNA-seq dataset from a KPY tumor organoid time course, but that is part of a separate manuscript (data not shown, *Moye et al., Embo J, 2024*). In our data from Moye et al, we could identify the two subsets as early as 4 days after activation of oncogenic Kras, persisting until the last time point we collected in that study at 14 days. It is noteworthy that the abundance of these subsets did change throughout the progression of tumorigenesis. In each study (Moye et al and here in Li et al, we further probed the importance of different targets in the oncogenic state that could be novel therapeutic targets. Furthermore, as the reviewer points out above, our findings correlate well with RNA and ATAC sequencing data from various time points in the published KP GEMM, supporting the idea that they are not relevant only for one snapshot of time in culture. Prior to this manuscript, it has not been shown in the previous publications that the cell states are similar in organoids or that they have a differential functional capacity for tumorigenesis.

Thirdly, as the reviewer suggested, we agreed it was important to examine the omics data in WT cells as a comparison. We have now compared cells from KP tumor organoids and WT organoids using the multi-omics methods (Please see revised Figure S1A and S1B). The KP and WT cells were distributed in separated clusters in UMAP, indicating dramatic transcriptional changes after oncogenic alteration. This also supports our conclusion that differences in clustering and cell states are not due to different levels of recombination.

2. Authors used Sftpc expression to classify Groups 1,2. However, data in Fig-1F shows Sftpc expression not much different in Group-2 compared to Group-3 and 4. Therefore, it's not clear why the authors grouped G1 and G2. Is G1 similar to unrecombined cells or WT cells? As mentioned in my comment #1, it is essential to directly compare current datasets with control cells.

Thanks for the nice suggestions. We grouped G1 and G2 for three reasons. Firstly, our differential gene analysis among the four groups of cells (Figure S1I) revealed a significant overlap in highly expressed genes between G1 and G2, indicating a notable similarity when compared to the other two groups. Notably,

the highly expressed genes in G1 and G2 demonstrate enrichment in Gene Ontology (GO) terms associated with alveolar lamellar body and surfactant homeostasis (Figure 1D and E). This enrichment suggests that these two groups of cells retain a portion of their original cellular function as AT2 cells, whose primary role in the normal lung is the synthesis and secretion of surfactant. Secondly, it is worth mentioning that both G1 and G2 still exhibit expression of the lung lineage regulator *Nkx2.1*, while G3 and G4 no longer express this gene. Numerous previous studies on KP lung adenocarcinoma provide substantial support for the notion that the loss of *NKX2.1* expression is a crucial step in the progression of tumors (Figure 1F). Lastly, when comparing our in vitro organoids dataset with the in vivo KY tumor dataset, we observed that G1 and G2 were consistently grouped together at both the gene expression level and chromatin accessibility level (Figure 1I and 1J). Rather than naming the groups AT2-like or EMT-like, we referred to them by SPC and *Hmga2* in order to keep nomenclature associated with gene expression of predominant markers that apply to the two major groups.

In an effort to enhance the robustness of our analysis, cells from control wild-type (WT) organoids have been incorporated. This inclusion has enabled us to affirm that G1 exhibits a greater similarity to recombined cells when contrasted with control cells, as delineated in the revised Figure S1A and S1B, presented above. While G1 demonstrates a heightened expression of *Sftpc* compared to G2, G3, and G4, the *Sftpc* expression level remains lower than that observed in control organoids.

3. Much of the data in this manuscript comes from correlation of gene expression and TF motif prediction. This manuscript does not provide any validation for these candidates using organoid models.

Thank you for the suggestion. Our integration analysis has revealed distinct transcription factor utilization in tumorigenesis between SPC-high and *Hmga2*-high cells. While many of the enriched transcription factors (TFs) have previously been rigorously validated for their significance in *Kras* mutant LUAD progression in vivo, our study highlights the cell state-specific roles of these TFs. For example, TFs highly expressed in the SPC-high cells and the *Hmga2*-high cells are required for tumorigenesis in the *Kras* model, such as *STAT3*, *Wnt*, and *NFκB* (Figure S2 A-C, D-F). We have now further validated the pro-tumorigenic properties of *Stat3* using our organoid system (Fig. S2 G and H). Cyto-spin staining revealed that whereas CD44-low and CD44-high cells exhibited comparable levels of *Stat3* protein, the CD44-low population displayed a higher percentage of phosphorylated *Stat3* protein, indicating elevated *Stat3* activity in the CD44-low population (Reviewer Figure 3A and B). Additionally, qPCR analysis indicated higher expression of putative *Stat3* target genes in CD44-low cells compared to CD44-high cells (Reviewer Figure 3C). Furthermore, we treated KPY tumor organoids with a *Stat3* inhibitor (*Stattic*) after plating for co-culture. The results demonstrated that 800 nM of the inhibitor treatment significantly reduced organoid growth (Figure S2 G and H) as well as the expression of putative *Stat3* target genes compared to the DMSO-treated conditions (Reviewer Figure 3D). These findings confirm the role of *Stat3* in promoting tumorigenesis via the CD44-low cell state. We are currently conducting experiments to validate other candidate TFs and their cell state specificity both in vitro and in vivo. However, the comprehensive validation process necessitates a considerable amount of time, which extends beyond the scope of this manuscript.

Figure for reviewers removed

4. Tumor progression: Does organoid model recapitulate tumorigenesis in vivo? For example, do they progress through AHA, adenoma, and adenocarcinoma similar to in vivo?

Yes, the organoid system models in vivo tumorigenesis including progression in stages. The progression of tumorigenic phenotypes in our Kras and Kras/p53 tumor organoids has been described in our previous publication, Dost et al 2020. (Reviewer Figure 4 and 5). There we noted that “Histological analysis revealed that our tumor organoid model recapitulated in vivo tumor progression.” Notably, H&E analysis was done over a time course. At days 7-10, Kras and Kras/p53 organoids demonstrated pleiomorphic nuclei similar to that observed in AHA and adenoma whereas wild-type organoid cells had normal nuclei. Giant multinucleated cells were observed at later time points in the tumor organoids, similar to those observed in KP adenocarcinomas in vivo at later stages of tumorigenesis.

In addition, upon transplantation via intratracheal administration in vivo, we observed the manifestation of various stages, including atypical hyperplasia (AHA), adenoma, and adenocarcinoma (Dost et al Figure 2(F–H) and this manuscript Figure 5C).

Figure for reviewers removed

5. In Fig3E-G, the authors show 3 types of organoids: SPChi, HMGA2hi, and mixed type. Are these differences due to cell of origin (i.e., subsets of AT2s) or different recombination events? Further, authors prior work has revealed that BASCs can generate mixed type (airway and alveolar) organoids. However, current study did not consider using different subsets of epithelial cells as starting population to assess any correlation between cell of origin and tumor organoid outcomes.

The concern regarding different recombination events has been addressed in response to reviewer question 1 above. Regarding the issue of a heterogeneous population with different subsets of AT2 cells being used to develop the tumor organoids, we cannot rule out this possibility formally. However, our data suggest this is not the cause of the different types of organoids that emerge after oncogenic Kras expression. First, to address reviewer questions above, we have added data from multi-omics analysis of wild-type alveolar organoids. Notably, the cells in wild-type organoids cluster into one major cluster in contrast to the KPY organoid cells. Whereas some possible subclusters can be seen within the wildtype organoids, they are not distinguishable by SPC or Hmga2 as are tumor organoids (please see Figure S1), and they do not overlap with cells from KPY organoids.

Indeed, our laboratory has also characterized the BASC population as a cell of origin of LUAD. However, for this manuscript and our initial publication describing tumor organoids, we focused only on the use of AT2 cells to study the transcriptional changes that occur early after Kras is activated for the very reason that the reviewer points out. We were interested in studying cellular state, and we did not want to confuse the results with the variety of epithelial cell types that the BASC population can generate, which includes

multipotent progenitors, AT2 cells and Club cells. It would be very interesting to compare tumor organoids generated from BASCs and AT2 cells, yet will require substantial work beyond this manuscript scope. We added this to the Limitations section of our revised manuscript.

6. The finding that SPChi cells have higher tumorigenic potential than HMGChi cells is interesting. However, its not clear what caused this? Is it due to differential engraftment of SPChi and HMGChi cells? Or is it due to positive or negative influence of support cells in the niche? Additionally, does the tumorigenic potential change in a different graft assay? For example, xenograft model or cells administered via tail vein.

We agree with the reviewer that SPChi cells have higher tumorigenic capacity than Hmga2hi cells is an interesting and unexpected finding and a highlight of our manuscript.

We have conducted additional in vivo transplantation experiments wherein recipient mice were sacrificed at earlier time points to assess potential differences in the ability of CD44-high and CD44-low cells to be transplanted and subsequently detected (referred to as engraftment). KP-Tomato organoids were grown and sorted on day 7 in culture based on CD44 expression. 50,000 cells per sample were transplanted into 4 recipient mice. Recipient mice were sacrificed after 2 weeks, lungs were digested, and flow cytometry analysis were performed to quantify the proportion of TdTomato+ cells out of total DAPI-/CD31-/CD45-/Epcam+ epithelial cells. There was no difference in the %EpCam+/TdTomato+ detected in mice that received CD44-Low and CD44-High KP organoid cells (Manuscript Figure S5C).

It is a very interesting question to ask if different subsets of KPY cells rely differently on supporting cells in the niche. Consistent with the in vivo transplantation result, our in vitro organoid-forming assay suggests that SPChi cells exhibit a greater advantage in the presence of lung stromal cells (Figure 4F and 4G). In addition, an increased proportion of stromal cells in the co-culture environment correlates with a higher percentage of SPC+ organoids (Figure S4D). Given these results, we agree with the reviewer that the difference in tumorigenic potential between the two cellular states may, at least in part, be influenced by the supportive effects of lung niche cells. More detailed understanding of this aspect of our study will require more experiments beyond the scope of this manuscript.

The alternative graft assays suggested by the reviewer, such as subcutaneous or tail vein injection assays, are commonly employed to investigate topics related to tumor progression and metastasis, respectively, and indeed assess tumorigenesis in a distinct niche. Whereas these are very interesting studies that we are planning to conduct in the future, we wished to keep this manuscript focused on the earliest stages of tumorigenesis within the lung.

Dear Dr Carla Kim,

Thank you for the submission of your amended manuscript (EMBOJ-2024-117108R) to The EMBO Journal, as well as for your patience with our response in light of the unusual protraction due to delayed expert input. We have carefully assessed your manuscript, and the point-by-point response provided to the referee concerns that were raised during review at a different journal. In addition, and as mentioned before, we decided to involve two arbitrating experts to evaluate the revised version of your work, with respect to technical robustness and overall suitability of your work for publication in The EMBO Journal.

As you will see from their comments enclosed below, the advisors are overall in favour of the work stating the interest and value of your results and they are supportive of publication at The EMBO Journal, pending satisfactory revision addressing the remaining minor issues indicated.

We are thus happy to invite a final revised version of your manuscript, with the following points to be addressed:

- > clarify remaining data questions (advisor #1's pt.5).
- > revisit data presentation and discussion (advisor #1's pts. 3,4).

Please note that we decided to editorially overrule the advisor #1's concerns regarding conceptual advance of the findings.

Please submit the revised manuscript using the link enclosed below.

Once we have received this amended version, we should then be able to swiftly complete formal acceptance and expedited production of the manuscript.

We also need you to take care of a number of minor issues related to formatting and data annotation, as detailed below.

Please let me know any time should you have questions regarding above points.

As you might remember from previous exchange, every paper at the EMBO Journal now includes a 'Synopsis', displayed on the html and freely accessible to all readers. The synopsis includes a 'model' figure as well as 2-5 one-short-sentence bullet points that summarize the article. I would appreciate if you could provide this figure and the bullet points.

Thank you again for giving us the chance to consider your manuscript for The EMBO Journal, I look forward to hearing from you and receiving your final complemented version of the manuscript.

Best regards,

Daniel Klimmeck

Daniel Klimmeck PhD
Senior Editor
The EMBO Journal.

>> Please add up to five keywords to your study.

>> Limit the abstract to maximally 175 words.

>> Author information: revisit e-mail contact provided in our system for co-author R.T. B. .

>> Author Contributions: Please remove the author contributions information from the manuscript text. Note that CRediT has replaced the traditional author contributions section as of now because it offers a systematic machine-readable author contributions format that allows for more effective research assessment. and use the free text boxes beneath each contributing author's name to add specific details on the author's contribution.

More information is available in our guide to authors.
<https://www.embopress.org/page/journal/14602075/authorguide>

>> Adjust the title of the 'Declaration of Interests' section to 'Disclosure and Competing Interests Statement' and move after Acknowledgements. Please add a statement: 'C.F.K. is a member of the EMBO Journal editorial advisory board.'

>> Section order should be corrected as follows: title page with complete author information, abstract, keywords, introduction, results, discussion, methods, data availability section, acknowledgements, disclosure and competing interests statement, references, main figure legends, tables, expanded figure legends.'

>> Figure callouts: the callouts for Fig S1 - 5 needs to be corrected to Fig EV1 - 5.

>> Please provide source data for the study as to the separate request e-mail by my colleague Hannah Sonntag.

>> Funding: please enter the following funding information in the list of funders in our online system: DRG-2523-24, F31 HL159919, Landry Cancer Consortium fellowship, LUNgevity Foundation, V Foundation for Cancer Research, the Thoracic Foundation, the Ellison Foundation, the American Lung Association LCD-619492 R35HL150876, P50CA265826, R01CA233671, The G. Harold & Leila Y. Mathers Foundation.

>> References: adjust reference format to EMBO Journal format, 10 authors et al, and place References after the Discussion, before figure legends.

>> Data availability section: provide a complete URL for the Zenodo dataset, and ensure the dataset is publicly accessible.

>> Add a Reagents and Tools table to the Methods section, listing key reagents, experimental models, software and relevant equipment.

>> Consider additional changes and comments from our production team as indicated below:

- DAS:

Please note that the specific URL for Zenodo dataset 10.5281/zenodo.7713052 is not provided in the data availability statement.

- Figure legends:

1. Please note that the exact p values are not provided in the legends of figures 4G, 5B, D; EV2 H
2. Please indicate the statistical test used for data analysis in the legends of figures 4G; 5B, D, E; EV2H; EV5 D
3. Please note that in figures EV2 H there is a mismatch between the annotated p values in the figure legend and the annotated p values in the figure file that should be corrected.
4. Please note that the box plots need to be defined in terms of minima, maxima, centre, bounds of box and whiskers, and percentile in the legends of figures 1F, 2D-F; 4B, EV2 A-F.
5. Please note that information related to n is missing in the legends of figures 1F, 2D-F; 3G, 4B, EV2 A-F; EV4 D, EV5 D.
6. Please note that the error bars are not defined in the legends of figures 3G, 4E, G ; 5B, D, E; EV2 H, EV4D, EV5 D.
7. Please note that for heatmap present in figures 2G-F a numbered scale bar is not provided. This needs to be rectified.
8. Please note that the scale bar needs to be defined for figure 4B.
9. Please note that scale bar and its definition are missing for figures 5C; EV4 C.

EMBOJ-2024-117108-T, arbitrating advisor #1's comment:

I went through the manuscript, and the authors' rebuttal to the reviewers' comments, and also looked at their earlier paper in which they also describe this LUAD organoid model (Dost et al 2020).

A few points:

- 1) One can indeed question what is new here (reviewer 2) although the story has more developed and now shows that SPC-positive KP cells act as a more potent tumor source than HMGA2 pos. cells. In Dost et al. they much more emphasized the importance of the progression of SPC-pos. cells towards more early development stages, in which HMGA2 can serve as a marker. Work from the Jack's lab has associated this embryonal phenotype with metastatic potential, an issue that is very relevant but not addressed in the current work. However, the story of the Kim group now emphasizes the more potent tumor growth of SPC positive KP cells. To illustrate this I quote from the discussion paragraph in Dost et al: "Furthermore, we see strong downregulation of the AT2 cell signature in murine and human organoids with almost complete loss of SPC expression (SPC pos) in murine organoids, whereas downregulation of the signature is more subtle in our GEMM data". A different message.
- 2) The authors emphasize that their work now shows that the EMT-like cells are less important than the field surmised (?) as this was according to the authors the reason there is focus on inhibiting these EMT stages as an intervention strategy. This seems primarily to be used as a selling point and is also beside the important role the more immature cells (e.g. HMGA2-pos) likely play in facilitating metastasis. In general, EMT is considered highly relevant for metastatic spread but is also considered a temporary - and important- transition stage with the reverse (MET) occurring upon the actual colonization of remote sites. A versatile switch between SPC positive KP cells and Hmga2 expressing KP cells can serve this role.
- 3) They do not test or discuss the possibility that the SPC-high and Hmga2-high cells might actually act in a paracrine fashion or that the embryonal stages (HMGA2-pos) play a role in metastasis. Their data almost scream to include experiments to test this. At least this should have been discussed in view of the substantial fraction of mixed tumors they observe after transplanting CD44 Low or CD44 High cells (fig. 5E). Such paracrine interaction is also noted in SCLC cells (neuroendocrine and non-neuroendocrine (the latter LUAD-like) populating the tumors.
- 4) The apparent easy conversion of the 2 overarching classes (Group 1-2 (1 and 2 appear actually quite different) and Group 3-4) should for clarity also be visualized in fig 5F in which they treat group 1-2 and group 3-4 as distinct lineages (now it is more an operational figure (like 1A) than an explanatory figure.
- 5) I was confused by the discussion of "Review Figure 3B" in their rebuttal. They claim STAT3 is equally expressed in CD44-high and CD44-low cells but that the signaling is evidently more significant in CD44-Low cells as only in the latter cells STAT3 is phosphorylated. However, in Fig 2G there is a pretty substantial difference in Stat3 RNA expression between these 2. It is uneasy if RNA and protein expression (and its phosphorylation) deviates that much without discussing this as one then wonders what the value is of scRNAseq (of stat3 and may be of other genes as well?) in such study to subdivide cell populations.

In conclusion: In my view they have properly addressed some of the points raised by the reviewers (e.g. partial recombination). But overall, I am lukewarm. The advance over Dost et al. is modest and the apparent divergence from this earlier work should have been explained or discussed (somewhat reminiscent of their BASC work decades ago), whereas at the same time a number of important questions have not been addressed.

EMBOJ-2024-117108-T, arbitrating advisor #2's comment:

Rev 1, pt1: The Reviewer raises an important point that the authors should address in either the results or the discussion along the lines of their response.

Rev , pt 6: Along with the new data the questions that the mechanistic concepts that the reviewer proposes, grafting vs niche effect, should be discussed in light of their data.

Generally, I am not sure why the new experimental results are back in the supplementary, not up front or Reviewer only and not included in the MS.

But overall, I think this would make a good contribution.

Response to *EMBO Journal* reviewer comments

We thank the EMBO Journal for this review and the referees for their additional points to improve our manuscript.

Reviewer comments:

EMBOJ-2024-117108-T, arbitrating advisor #1's comments:

1) One can indeed question what is new here (reviewer 2) although the story has more developed and now shows that SPC-positive KP cells act as a more potent tumor source than HMGA2 pos. cells. In Dost et al. they much more emphasized the importance of the progression of SPC-pos. cells towards more early development stages, in which HMGA2 can serve as a marker. Work from the Jack's lab has associated this embryonal phenotype with metastatic potential, an issue that is very relevant but not addressed in the current work. However, the story of the Kim group now emphasizes the more potent tumor growth of SPC positive KP cells. To illustrate this, I quote from the discussion paragraph in Dost et al: "Furthermore, we see strong downregulation of the AT2 cell signature in murine and human organoids with almost complete loss of SPC expression (SPC pos) in murine organoids, whereas downregulation of the signature is more subtle in our GEMM data". A different message.

Response: In this manuscript, we indeed wish to highlight the finding that SPC-high cells, which more closely resemble AT2 cells in gene expression, have higher tumorigenic potential in the early stages of KP tumorigenesis. Our previous work (Dost et al., 2020) was focused on the discovery that at early stages, there are cells present that have lost their AT2 identity. Although we emphasized the discovery of the cells with loss of AT2 identity, we did not test their relative function in tumorigenesis compared to other cell states, which we begin to do here in this manuscript. In addition, the scRNA-seq data in Dost et al. was from organoids and the GEMM with Kras-G12D and wildtype p53, whereas this paper used the model with p53 loss. We have edited the discussion to better reflect these points.

2) The authors emphasize that their work now shows that the EMT-like cells are less important than the field surmised (?) as this was according to the authors the reason there is focus on inhibiting these EMT stages as an intervention strategy. This seems primarily to be used as a selling point and is also beside the important role the more immature cells (e.g. HMGA2-pos) likely play in facilitating metastasis. In general, EMT is considered highly relevant for metastatic spread but is also considered a temporary -and important- transition stage with the reverse (MET) occurring upon the actual colonization of remote sites. A versatile switch between SPC positive KP cells and Hmga2 expressing KP cells can serve this role.

Response: We agree that it will be important to determine the potential of SPC-high cells to switch to HMGA2-high cells in future studies. We do not wish to interpret that the EMT-like cells are not important, but rather that they have comparatively less tumorigenic capacity at the time point examined in our studies.

3) They do not test or discuss the possibility that the SPC-high and Hmga2-high cells might actually act in a paracrine fashion or that the embryonal stages (HMGA2-pos) play a role in metastasis. Their data almost scream to include experiments to test this. At least this should have been discussed in view of the substantial fraction of mixed tumors they observe after transplanting CD44 Low or CD44 High cells (fig. 5E). Such paracrine interaction is also noted in SCLC cells (neuroendocrine and non-neuroendocrine (the latter LUAD-like) populating the tumors.

Response: We agree that it will be important to determine the potential of SPC-high cells to switch to HMGA2-high cells and contribute to metastasis either from a direct role or via paracrine signaling. However, this will require additional experimentation beyond this manuscript. We have added more discussion on these topics to the manuscript Discussion section.

4) The apparent easy conversion of the 2 overarching classes (Group 1-2 (1 and 2 appear actually quite different) and Group 3-4) should for clarity also be visualized in fig 5F in which they treat group 1-2 and group 3-4 as distinct lineages (now it is more an operational figure (like 1A) than an explanatory figure.

Response: We have added bidirectional arrows between the two groups of cells in Figure 5F to reflect the potential ability of the cells states to switch.

5) I was confused by the discussion of "Review Figure 3B" in their rebuttal. They claim STAT3 is equally expressed in CD44-high and CD44-low cells but that the signaling is evidently more significant in CD44-Low cells as only in the latter cells STAT3 is phosphorylated. However, in Fig 2G there is a pretty substantial difference in Stat3 RNA expression between these 2. It is uneasy if RNA and protein expression (and its phosphorylation) deviates that much without discussing this as one then wonders what the value is of scRNAseq (of stat3 and may be of other genes as well?) in such study to subdivide cell populations.

Response: We acknowledge the potential confusion between RNA expression levels and protein levels of STAT3 in our data comparing expression, ATAC and immunostaining results.

It worth noting that gene expression alone may not always be sufficient to accurately predict the activity of certain genes, particularly transcription factors, which are often heavily regulated at the post-transcriptional level. To address this, we employed regulon analysis to better understand the regulatory role of STAT3 in the SPC-high population. However, for other genes that participate in specific cellular activities, such as SFTPC (surfactant protein), gene expression is still closely linked to their function.

First, as the reviewer noted, as shown in the heat maps (previous Figure 2G and revised Figure 2D), the RNA-seq from our multi-omics study shows that STAT3 expression is differential between the Groups. The heat maps show STAT3 expression is highest in Group 1 cells in comparison to Group 2, 3 and 4 (revised 2D). Importantly, the heat maps display the data taking into account the entire gene expression program. The raw data for STAT3 are represented in revised Figure 2E, where it can be appreciated that Group 1 cells exhibit the highest STAT3 expression, with Groups 2, 3 and 4 showing similar levels of STAT3. The UMAP in Figure 2E shows that all the Groups have cells that express STAT3. The enrichment of the STAT3 motif is also similar across the Groups in 2F. In our view, the most notable difference can be seen in Figure 2G, which shows expression of STAT3 regulon, the STAT3 target genes, highest in Group 1 compared to Group 2, higher in Group 2 versus Group 3, and higher in Group 3 versus Group 4.

In order to validate these RNA and ATAC seq results, and to perform functional analysis of the cell Groups, we made use of the surface marker CD44. It is important to note that by selecting cells that are CD44-low, we have a cell population with both Group 1 and Group 2 cells, allowing us to test SPC-high cells. Similarly, selection of CD44-high cells includes Group 3 and Group 4 cells, allowing us to test Hmga2-high cells. Thus, gene expression differences that are significantly different in Group 1 versus Group 2 cannot be appreciated; the CD44 strategy only makes it possible to compare Group1+2 versus Group 3+4. This means that expression differences in some genes of interest, such as STAT3, may be less apparent when comparing CD44-low and CD44-high cells.

The results in our further experimentation comparing CD44-low and CD44-high cells supports these interpretations. In our additional studies, organoid cells were sorted by CD44 levels (low or high) to obtain cells that approximately correspond to SPC-high and HMGA2-high cell states, respectively. Fitting with our hypothesis that STAT3 activity is substantially different between the cell states, we detected significantly more expression of Abca3 and Etv5, two known STAT3 target genes, in CD44-low cells compared to CD44-high cells (Figure 4J). Whereas both CD44-low and CD44-high populations contained cells with STAT3 detected by IF, there were significantly more cells with phosphorylated STAT3 protein in the SPC-high cells (CD44-low) (Figure 4J). STAT3 phosphorylation is associated with its function as a transcription activator. Thus, we conclude that STAT3 activity has a more critical role in SPC-high cells than HMGA2-high cells, and that the RNA-seq studies are still valuable to identify candidate genes of interest based on gene expression.

EMBOJ-2024-117108-T, arbitrating advisor #2's comments:

Rev 1, pt1

The Reviewer raises an important point that the authors should address in either the results or the discussion along the lines of their response.

Response: We agree that the main emphasis of this manuscript is the important functional role of the SPC-high cell state in the early stages of tumorigenesis, and we have added discussion to highlight this point. The revised discussion also better compares this manuscript to our previous work.

Rev , pt 6

Along with the new data the questions that the mechanistic concepts that the reviewer proposes, grafting vs niche effect, should be discussed in light of their data.

Response: We have added discussion points to the revised manuscript regarding the potential role of the cell states in metastasis, paracrine signaling and dependence on niche.

Generally, I am not sure why the new experimental results are back in the supplementary, not up front or Reviewer only and not included in the MS.

Response: We appreciate the reviewer's comment, and as a result we have moved new experimental results related to STAT3 into the manuscript main figures and/or EV. We also updated the previous Response to reviewers document to reflect the changes in data location in the manuscript.

Dear Dr Carla Kim, dear Dr Jingyun Li,

Thank you for submitting the revised version of your manuscript. I have now evaluated your amended manuscript and concluded that the remaining minor concerns have been sufficiently addressed.

I am thus pleased to inform you that your manuscript has been accepted for publication in the EMBO Journal.

Please note that it is The EMBO Journal policy for the transcript of the editorial process (containing referee reports and your response letter) to be published as an online supplement to each paper. More information is available here: https://www.embopress.org/transparent-process#Review_Process

On a different note, I would like to alert you that EMBO Press offers a format for a video-synopsis of work published with us, which essentially is a short, author-generated film explaining the core findings in hand drawings, and, as we believe, can be very useful to increase visibility of the work. Please see the following link for representative examples and their integration into the article web page:

<https://www.embopress.org/doi/full/10.15252/emj.2019103932>

Best regards,

Daniel Klimmeck

Daniel Klimmeck, PhD
Senior Editor
The EMBO Journal
EMBO
Postfach 1022-40
Meyerhofstrasse 1
D-69117 Heidelberg
contact@embojournal.org